# FedBN: Federated Learning on Non-IID Features via Local Batch Normalization

**Xiaoxiao Li**
Department of Computer Science
Princeton University
xiaoxiao.li@aya.yale.edu

**Meirui Jiang**
Department of Computer Science and Engineering
The Chinese University of Hong Kong
mrjiang@cse.cuhk.edu.hk

**Xiaofei Zhang**
Department of Statistics
Iowa State University
xfzhang@iastate.edu

**Michael Kamp**
Dept of Data Science and AI, Faculty of IT
Monash University
michael.kamp@monash.edu

**Qi Dou**[*]
Department of Computer Science and Engineering
The Chinese University of Hong Kong
qdou@cse.cuhk.edu.hk

## Abstract

The emerging paradigm of federated learning (FL) strives to enable collaborative training of deep models on the network edge without centrally aggregating raw data and hence improving data privacy. In most cases, the assumption of independent and identically distributed samples across local clients does not hold for federated learning setups. Under this setting, neural network training performance may vary significantly according to the data distribution and even hurt training convergence. Most of the previous work has focused on a difference in the distribution of labels or client shifts. Unlike those settings, we address an important problem of FL, e.g., different scanners/sensors in medical imaging, different scenery distribution in autonomous driving (highway vs. city), where local clients store examples with different distributions compared to other clients, which we denote as *feature shift non-iid*. In this work, we propose an effective method that uses local batch normalization to alleviate the feature shift before averaging models. The resulting scheme, called *FedBN*, outperforms both classical FedAvg, as well as the state-of-the-art for non-iid data (FedProx) on our extensive experiments. These empirical results are supported by a convergence analysis that shows in a simplified setting that FedBN has a faster convergence rate than FedAvg. Code is available at https://github.com/med-air/FedBN.

## 1 Introduction

Federated learning (FL), has gained popularity for various applications involving learning from distributed data. In FL, a cloud server (the "server") can communicate with distributed data sources (the "clients"), while the clients hold data separately. A major challenge in FL is the training data statistical heterogeneity among the clients (Kairouz et al., 2019; Li et al., 2020b). It has been shown that standard federated methods such as FedAvg (McMahan et al., 2017) which are not designed particularly taking care of non-iid data significantly suffer from performance degradation or even diverge if deployed over non-iid samples (Karimireddy et al., 2019; Li et al., 2018; 2020a).

Recent studies have attempted to address the problem of FL on non-iid data. Most variants of FedAvg primarily tackle the issues of stability, client drift and heterogeneous label distribution over clients (Li et al., 2020b; Karimireddy et al., 2019; Zhao et al., 2018). Instead, we focus on the shift

---

[*]Corresponding author.

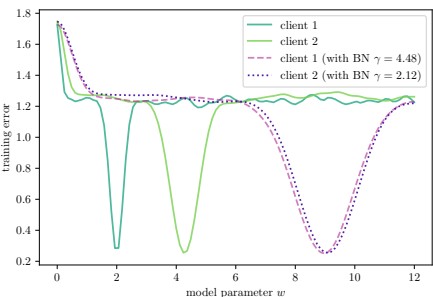
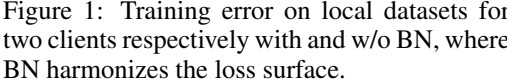
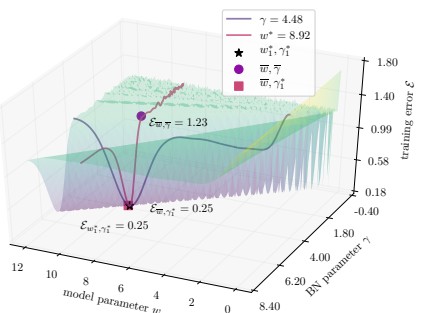

Figure 1: Training error on local datasets for two clients respectively with and w/o BN, where BN harmonizes the loss surface.

Figure 2: Error surface of a client for model parameter $w \in [0.001, 12]$ and BN parameter $\gamma \in [0.001, 4]$. Averaging model *and* BN parameters leads to worse solutions.

in the feature space, which has not yet been explored in the literature. Specifically, we consider that local data deviates in terms of the distribution in feature space, and identify this scenario as *feature shift*. This type of non-iid data is a critical problem in many real-world scenarios, typically in cases where the local devices are responsible for a heterogeneity in the feature distributions. For example in cancer diagnosis tasks, medical radiology images collected in different hospitals have uniformly distributed labels (i.e., the cancer types treated are quite similar across the hospitals). However, the image appearance can vary a lot due to different imaging machines and protocols used in hospitals, e.g., different intensity and contrast. In this example, each hospital is a client and hospitals aim to collaboratively train a cancer detection model without sharing privacy-sensitive data.

Tackling non-iid data with feature shift has been explored in classical centralized training in the context of domain adaptation. Here, an effective approach in practice is utilizing Batch Normalization (BN) (Ioffe & Szegedy, 2015): recent work has proposed BN as a tool to mitigate domain shifts in domain adaptation tasks with promising results achieved (Li et al., 2016; Liu et al., 2020; Chang et al., 2019). Inspired by this, this paper proposes to apply BN for feature shift FL. To illustrate the idea, we present a toy example that illustrates how BN may help harmonizing local feature distributions.

**Observation of BN in a FL Toy Example:** We consider a simple non-convex learning problem: we generate data $x, y \in \mathbb{R}$ with $y = \cos(w_{true}x) + \epsilon$, where $x \in \mathbb{R}$ is drawn iid from Gaussian distribution and $\epsilon$ is zero-mean Gaussian noise and consider models of the form $f_w(x) = \cos(wx)$ with model parameter $w \in \mathbb{R}$. Local data deviates in the variance of $x$. First, we illustrate that local batch normalization harmonizes local data distributions. We consider a simplified form of BN that normalizes the input by scaling it with $\gamma$, i.e., the local empirical standard deviation, and a setting with 2 clients. As Fig. 1 shows, the local squared loss is very different between the two clients. Thus, averaging the model does not lead to a good model. However when applying local BN, the local training error surfaces become similar and averaging the models can be beneficial. To further illustrate the impact of BN, we plot the error surface for one client with respect to both model parameter $w \in \mathbb{R}$ and BN parameter $\gamma \in \mathbb{R}$ in Fig. 2. The figure shows that for an optimal weight $w_1^*$, changing $\gamma$ deteriorates the model quality. Similarly, for a given optimal BN parameter $\gamma_1^*$, changing $w$ deteriorates the quality. In particular, the average model $\overline{w} = (w_1^* + w_2^*)/2$ and average BN parameters $\overline{\gamma} = (\gamma_1^* + \gamma_2^*)/2$ has a high generalization error. At the same time, the average model $\overline{w}$ with local BN parameter $\gamma_1^*$ performs very well.

Motivated by the above insight and observation, this paper proposes a novel federated learning method, called FedBN, for addressing non-iid training data which keeps the client BN layers updated locally, without communicating, and aggregating them at the server. In practice, we can simply update the non-BN layers using FedAvg, without modifying any optimization or aggregation scheme. This approach has zero parameters to tune, requires minimal additional computational resources, and can be easily applied to arbitrary neural network architectures with BN layers in FL. Besides the benefit shown in the toy example, we also show the benefits in accelerating convergence by theoretically analyzing the convergence of FedBN in the over-parameterized regime. In addition,

we have conducted extensive experiments on a benchmark and three real-world datasets. Compared to classical FedAvg, as well as the state-of-the-art for non-iid data (FedProx), our novel method, FedBN, demonstrates significant practical improvements on the extensive experiments.

## 2 RELATED WORK

**Techniques for Non-IID Challenges in Federated Learning:**  The widely known aggregation strategy in FL, FedAvg (McMahan et al., 2017), often suffers when data is heterogeneous over local client. Empirical work addressing non-iid issues, mainly focus on label distribution skew, where a non-iid dataset is formed by partitioning a "flat" existing dataset based on the labels. FedProx (Li et al., 2020b), a recent framework tackled the heterogeneity by allowing partial information aggregation and adding a proximal term to FedAvg. Zhao et al. (2018) assumed a subset of the data is globally shared between all the clients, hence generalizes to the problem at hand. FedMA (Wang et al., 2020) proposed an aggregation strategy for non-iid data partition that shares global model in a layer-wise manner. However, so far there are only limited attempts considering non-iid induced from feature shift, which is common in medical data collecting from different equipment and natural image collected in various noisy environment. Very recently, FedRobust (Reisizadeh et al., 2020) assumes data follows an affine distribution shift and tackles this problem by learning the affine transformation. This hampers the generalization when we cannot estimate the explicit affine transformation. Concurrently to our work, SiloBN Andreux et al. (2020) empirically shows that local clients keeping some untrainable BN parameters could improve robustness to data heterogeneity, but provides no theoretical analysis of the approach. FedBN instead keeps all BN parameters strictly local. Recently, an orthogonal approach to the non-iid problem has been proposed that focuses on improving the optimization mechanism (Reddi et al., 2020; Zhang et al., 2020).

**Batch Normalization in Deep Neural Networks:**  Batch Normalization (Ioffe & Szegedy, 2015) is an indispensable component in many deep neural networks and has shown its success in neural network training. Relevant literature has uncovered a number of benefits given by batch normalization. Santurkar et al. (2018) showed that BN makes the optimization landscape significantly smoother. Luo et al. (2018) investigated an explicit regularization form of BN such that improving the robustness of optimization. Morcos et al. (2018) suggested that BN implicitly discourages single direction reliance, thus improving model generalizability. Li et al. (2018) took advantage of BN for tackling the domain adaptation problem. However, what a role BN is playing in the scope of federated learning, especially for non-iid training, still remains unexplored to date.

## 3 PRELIMINARY

**Non-IID Data in Federated Learning:** We introduce the concept of *feature shift* in federated learning as a novel category of client's non-iid data distribution. So far, the categories of non-iid data considered according to Kairouz et al. (2019); Hsieh et al. (2019) can be described by the joint probability between features $\mathbf{x}$ and labels $y$ on each client. We can rewrite $P_i(\mathbf{x}, y)$ as $P_i(y|\mathbf{x})P_i(\mathbf{x})$ and $P_i(\mathbf{x}|y)P_i(y)$. We define *feature shift* as the case that covers: 1) *covariate shift*: the marginal distributions $P_i(\mathbf{x})$ varies across clients, even if $P_i(y|\mathbf{x})$ is the same for all client; and 2) *concept shift*: the conditional distribution $P_i(\mathbf{x}|y)$ varies across clients and $P(y)$ is the same.

**Federated Averaging (FedAvg):** We establish our algorithm on FedAvg introduced by McMahan et al. (2017) which is the most popular existing and easiest to implement federated learning strategy, where clients collaboratively send updates of locally trained models to a global server. Each client runs a local copy of the global model on its local data. The global model's weights are then updated with an average of local clients' updates and deployed back to the clients. This builds upon previous distributed learning work by not only supplying local models but also performing training locally on each device. Hence FedAvg potentially empowers clients (especially clients with small dataset) to collaboratively learn a shared prediction model while keeping all training data locally. Although FedAvg has shown successes in classical Federated Learning tasks, it suffers from slow convergence and low accuracy in most non-iid contents (Li et al., 2020b; 2019).

## 4    FEDERATED AVERAGING WITH LOCAL BATCH NORMALIZATION

### 4.1    PROPOSED METHOD - FEDBN

We propose an efficient and effective learning strategy denoted *FedBN*. Similar to FedAvg, FedBN performs local updates and averages local models. However, FedBN assumes local models have BN layers and excludes their parameters from the averaging step. We present the full algorithm in Appendix C. This simple modification results in significant empirical improvements in non-iid settings. We provide an explanation for these improvements in a simplified scenario, in which we show that FedBN improves the convergence rate under feature shift.

### 4.2    PROBLEM SETUP

We assume $N \in \mathbb{N}$ clients to jointly train for $T \in \mathbb{N}$ epochs and to communicate after $E \in \mathbb{N}$ local iterations. Thus, the system has $T/E$ communication rounds over the $T$ epochs. For simplicity, we assume all clients to have $M \in \mathbb{N}$ training examples (a difference in training examples can be account for by weighted averaging (McMahan et al., 2017)) for a regression task, i.e., each client $i \in [N]$ ($[N] = \{1, \ldots, N\}$) has training examples $\{(\mathbf{x}_j^i, y_j^i) \in \mathbb{R}^d \times \mathbb{R} : j \in [M]\}$. Furthermore, we assume a two-layer neural network with ReLU activations trained by gradient descent. Let $\mathbf{v}_k \in \mathbb{R}^d$ denote the parameters of the first layer, where $k \in [m]$ and $m$ is the width of the hidden layer. Let $\| \mathbf{v} \|_{\mathbf{S}} \triangleq \sqrt{\mathbf{v}^\top \mathbf{S} \mathbf{v}}$ denote the induced vector norm for a positive definite matrix $\mathbf{S}$.

We consider a non-iid setting in FL where local feature distributions differ—not label distribution, as considered, e.g., in McMahan et al. (2017); Li et al. (2019). To be more precise, we make the following assumption.

**Assumption 4.1** (Data Distribution). *For each client $i \in [N]$ the inputs $\mathbf{x}_j^i$ are centered ($\mathbb{E}\mathbf{x}^i = \mathbf{0}$) with covariance matrix $\mathbf{S}_i = \mathbb{E}\mathbf{x}^i\mathbf{x}^{i\top}$, where $\mathbf{S}_i$ is independent from the label $\mathbf{y}$ and may differ for each $i \in [N]$ e.g., $\mathbf{S}_i$ are not all identity matrices, and for each index pair $p \neq q$, $\mathbf{x}_p \neq \kappa \cdot \mathbf{x}_q$ for all $\kappa \in \mathbb{R} \setminus \{0\}$.*

With Assumption 4.1, the normalization of the first layer for client $i$ is $\frac{\mathbf{v}_k^\top \mathbf{x}^i}{\|\mathbf{v}_k\|_{\mathbf{S}_i}}$. FedBN with client-specified BN parameters trains a model $f^* : \mathbb{R}^d \to \mathbb{R}$ parameterized by $(\mathbf{V}, \gamma, \mathbf{c}) \in \mathbb{R}^{m \times d} \times \mathbb{R}^{m \times N} \times \mathbb{R}^m$, i.e.,

$$f^*(\mathbf{x}; \mathbf{V}, \gamma, \mathbf{c}) = \frac{1}{\sqrt{m}} \sum_{k=1}^m c_k \sum_{i=1}^N \sigma \left( \gamma_{k,i} \cdot \frac{\mathbf{v}_k^\top \mathbf{x}}{\| \mathbf{v}_k \|_{\mathbf{S}_i}} \right) \cdot \mathbb{1}\{\mathbf{x} \in \text{client } i\} \ , \tag{1}$$

where $\gamma$ is the scaling parameter of BN and $\sigma(s) = \max\{s, 0\}$ is the ReLU activation function, $\mathbf{c}$ is the top layer parameters of the network. Here, we omit learning the shift parameter of BN [1]. FedAvg instead trains a function $f : \mathbb{R}^d \to \mathbb{R}$ which is a special case of Eq. 1 with $\gamma_{k,i} = \gamma_k$ for $\forall i \in [N]$. We take a random initialization of the parameters (Salimans & Kingma, 2016) in our analysis:

$$\mathbf{v}_k(0) \sim N\left(0, \alpha^2 \mathbf{I}\right), \quad c_k \sim U\{-1, 1\}, \quad \text{and} \quad \gamma_k = \gamma_{k,i} = \|\mathbf{v}_k(0)\|_2 / \alpha, \tag{2}$$

where $\alpha^2$ controls the magnitude of $\mathbf{v}_k$ at initialization. The initialization of the BN parameters $\gamma_k$ and $\gamma_{k,i}$ are independent of $\alpha$. The parameters of the network $f^*(\mathbf{x}; \mathbf{V}, \gamma, \mathbf{c})$ are obtained by minimizing the empirical risk with respect to the squared loss using gradient descent :

$$L(f^*) = \frac{1}{NM} \sum_{i=1}^N \sum_{j=1}^M \left( f^*(\mathbf{x}_j^i) - y_j^i \right)^2 \ . \tag{3}$$

### 4.3    CONVERGENCE ANALYSIS

Here we study the trajectory of networks FedAvg ($f$) and FedBN ($f^*$)'s prediction through the neural tangent kernel (NTK) introduced by Jacot et al. (2018). Recent machine learning theory

---

[1]We omit centering neurons as well as learning the shift parameter of BN for the neural network analysis because of the assumption that $\mathbf{x}$ is zero-mean and the two layer network setting (Kohler et al., 2019; Salimans & Kingma, 2016).

studies (Arora et al., 2019; Du et al., 2018; Allen-Zhu et al., 2019; van den Brand et al., 2020; Dukler et al., 2020) have shown that for finite-width over-parameterized networks, the convergence rate is controlled by the least eigenvalue of the induced kernel in the training evolution.

To simplify tracing the optimization dynamics, we consider the case that the number of local updates $E$ is 1. We can decompose the NTK into a magnitude component $\mathbf{G}(t)$ and direction component $\mathbf{V}(t)/\alpha^2$ following Dukler et al. (2020):

$$\frac{d\mathbf{f}}{dt} = -\mathbf{\Lambda}(t)(\mathbf{f}(t) - \mathbf{y}), \quad \text{where} \quad \mathbf{\Lambda}(t) := \frac{\mathbf{V}(t)}{\alpha^2} + \mathbf{G}(t).$$

The specific forms of $\mathbf{V}(t)$ and $\mathbf{G}(t)$ are given in Appendix B.1. Let $\lambda_{min}(A)$ denote the minimal eigenvalue of matrix $A$. The matrices $\mathbf{V}(t)$ and $\mathbf{G}(t)$ are positive semi-definite, since they can be viewed as covariance matrices. This gives $\lambda_{\min}(\mathbf{\Lambda}(t)) \geq \max\left\{\lambda_{\min}(\mathbf{V}(t))/\alpha^2, \lambda_{\min}(\mathbf{G}(t))\right\}$. According to NTK, the convergence rate is controlled by $\lambda_{min}(\mathbf{\Lambda}(t))$. Then, for $\alpha > 1$, convergence is dominated by $\mathbf{G}(t)$. Let $\mathbf{\Lambda}(t)$ and $\mathbf{\Lambda}^*(t)$ denote the evolution dynamics of FedAvg and FedBN and let $\mathbf{G}(t)$ and $\mathbf{G}^*(t)$ denote the magnitude component in the evolution dynamics of FedAvg and FedBN. For the convergence analysis, we use the auxiliary version of the Gram matrices, which is defined as follows.

**Definition 4.2.** *Given sample points $\{\mathbf{x}_p\}_{p=1}^{NM}$, we define the auxiliary Gram matrices $\mathbf{G}^\infty \in \mathbb{R}^{NM \times NM}$ and $\mathbf{G}^{*\infty} \in \mathbb{R}^{NM \times NM}$ as*

$$\mathbf{G}_{pq}^\infty := \mathbb{E}_{\mathbf{v} \sim N(0,\alpha^2\mathbf{I})} \sigma\left(\mathbf{v}^\top \mathbf{x}_p\right) \sigma\left(\mathbf{v}^\top \mathbf{x}_q\right), \quad \textit{(FedAvg)} \tag{4}$$

$$\mathbf{G}_{pq}^{*\infty} := \mathbb{E}_{\mathbf{v} \sim N(0,\alpha^2\mathbf{I})} \sigma\left(\mathbf{v}^\top \mathbf{x}_p\right) \sigma\left(\mathbf{v}^\top \mathbf{x}_q\right) \mathbb{1}\{i_p = i_q\}, \quad \textit{(FedBN)}. \tag{5}$$

Given Assumption 4.1, we use the key results in Dukler et al. (2020) to show that $\mathbf{G}^\infty$ is positive definite. Further, we show that $\mathbf{G}^{*\infty}$ is positive definite. We use the fact that the distance between $\mathbf{G}(t)$ and its auxiliary version is small in over-parameterized neural network, such that $\mathbf{G}(t)$ remains positive definite.

**Lemma 4.3.** *Fix points $\{\mathbf{x}_p\}_{p=1}^{NM}$ satisfying Assumption 4.1. Then Gram matrices $\mathbf{G}^\infty$ and $\mathbf{G}^{*\infty}$ defined as in (4) and (5) are strictly positive definite. Let the least eigenvalues be $\lambda_{\min}(\mathbf{G}^\infty) =: \mu_0$ and $\lambda_{\min}(\mathbf{G}^{*\infty}) =: \mu_0^*$, where $\mu_0, \mu_0^* > 0$.*

**Proof sketch** The main idea follows Du et al. (2018); Dukler et al. (2020), that given points $\{\mathbf{x}_p\}_{p=1}^{NM}$, the matrices $\mathbf{G}^\infty$ and $\mathbf{G}^{*\infty}$ can be shown as covariance matrix of linearly independent operators. More details of the proof are given in the Appendix B.2.

Based on our formulation, the convergence rate of FedAvg (Theorem 4.4) can be derived from Dukler et al. (2020) by considering non-identical covariance matrices. We derive the convergence rate of FedBN in Corollary 4.5. Our key result of comparing the convergence rates between FedAvg and FedBN is culminated in Corollary 4.6.

**Theorem 4.4** (G-dominated convergence for FedAvg Dukler et al. (2020)). *Suppose network (4) is initialized as in (2) with $\alpha > 1$, trained using gradient descent and Assumptions 4.1 holds. Given the loss function of training the neural network is the square loss with targets $\mathbf{y}$ satisfying $\|\mathbf{y}\|_\infty = O(1)$. If $m = \Omega\left(\max\left\{N^4M^4 \log(NM/\delta)/\alpha^4\mu_0^4, N^2M^2 \log(NM/\delta)/\mu_0^2\right\}\right)$, then with probability $1 - \delta$,*

1. *For iterations $t = 0, 1, \cdots$, the evolution matrix $\mathbf{\Lambda}(t)$ satisfies $\lambda_{\min}(\mathbf{\Lambda}(t)) \geq \frac{\mu_0}{2}$.*

2. *Training with gradient descent of step-size $\eta = O\left(\frac{1}{\|\mathbf{\Lambda}(t)\|}\right)$ converges linearly as*

$$\|\mathbf{f}(t) - \mathbf{y}\|_2^2 \leq \left(1 - \frac{\eta\mu_0}{2}\right)^t \|\mathbf{f}(0) - \mathbf{y}\|_2^2.$$

Following the key ideas in Dukler et al. (2020), here we further characterize the convergence for FedBN.

**Corollary 4.5** (G-dominated convergence for FedBN). *Suppose network (5) and all other conditions in Theorem 4.4. With probability $1 - \delta$, for iterations $t = 0, 1, \cdots$, the evolution matrix $\mathbf{\Lambda}^*(t)$ satisfies $\lambda_{\min}(\mathbf{\Lambda}^*(t)) \geq \frac{\mu_0^*}{2}$ and training with gradient descent of step-size $\eta = O\left(\frac{1}{\|\mathbf{\Lambda}^*(t)\|}\right)$ converges linearly as $\|\mathbf{f}^*(t) - \mathbf{y}\|_2^2 \leq \left(1 - \frac{\eta\mu_0^*}{2}\right)^t \|\mathbf{f}^*(0) - \mathbf{y}\|_2^2$.*

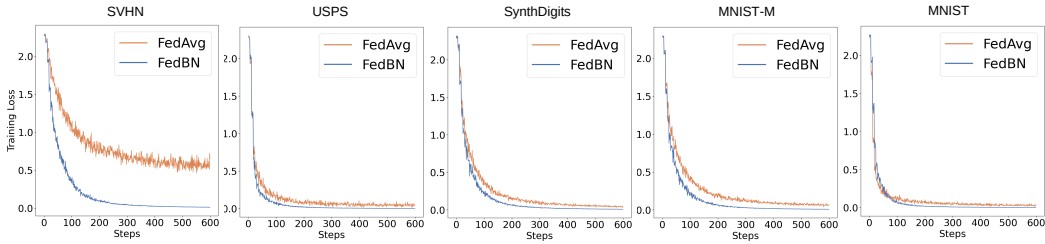

Figure 3: Convergence of the training loss of FedBN and FedAvg on the digits classification datasets. FedBN exhibits faster and more robust convergence.

The exponential factor of convergence for FedAvg $(1 - \eta\mu_0/2)$ and FedBN $(1 - \eta\mu_0^*/2)$ are controlled by the smallest eigenvalue of $\mathbf{G}(t)$, respectively $\mathbf{G}^*(t)$. Then we can analyze the convergence performance of FedAvg and FedBN by comparing $\lambda_{\min}(\mathbf{G}^\infty)$ and $\lambda_{\min}(\mathbf{G}^{*\infty})$.

**Corollary 4.6** (Convergence rate comparison between FedAvg and FedBN). *For the $\mathbf{G}$-dominated convergence, the convergence rate of FedBN is faster than that of FedAvg.*

**Proof sketch**   The key is to show $\lambda_{\min}(\mathbf{G}^\infty) \leq \lambda_{\min}(\mathbf{G}^{*\infty})$. Comparing equation (4) and (5), $\mathbf{G}^{*\infty}$ takes the $M \times M$ block matrices on the diagonal of $\mathbf{G}^\infty$. Let $\mathbf{G}_i^\infty$ be the $i$-th $M \times M$ block matrices on the diagonal of $\mathbf{G}^\infty$. By linear algebra, $\lambda_{\min}(\mathbf{G}_i^\infty) \geq \lambda_{\min}(\mathbf{G}^\infty)$ for $i \in [N]$. Since $\mathbf{G}^{*\infty} = diag(\mathbf{G}_1^\infty, \cdots, \mathbf{G}_N^\infty)$, we have $\lambda_{\min}(\mathbf{G}^{*\infty}) = \min_{i \in [N]}\{\lambda_{\min}(\mathbf{G}_i^\infty)\}$. Therefore, we have the result $\lambda_{\min}(\mathbf{G}^{*\infty}) \geq \lambda_{\min}(\mathbf{G}^\infty)$.

## 5   EXPERIMENTS

In this section, we demonstrate that using local BN parameters is beneficial in the presence of feature shift across clients with heterogeneity data. Our novel local parameter sharing strategy, FedBN, achieves more robust and faster convergence for feature shift non-iid datasets and obtains better model performance compared to alternative methods. This is shown on both benchmark and large real-world datasets.

### 5.1   BENCHMARK EXPERIMENTS

**Settings:**   We perform an extensive empirical analysis using a benchmark digits classification task containing different data sources with feature shift where each dataset is from a different domain. Data of different domains have heterogeneous appearance but share the same labels and label distribution. Specifically, we use the following five datasets: SVHN Netzer et al. (2011), USPS Hull (1994), SynthDigits Ganin & Lempitsky (2015), MNIST-M Ganin & Lempitsky (2015) and MNIST LeCun et al. (1998). To match the setup in Section 4, we truncate the sample size of the five datasets to their smallest number with random sampling, resulting in 7438 training samples in each dataset [2]. Testing samples are held out and kept the same for all the experiments on this benchmark dataset. Our classification model is a convolutional neural network where BN layers are added following each feature extraction layer (i.e., both convolutional and fully-connected). The architecture is detailed in Appendix D.2. For model training, we use the cross-entropy loss and SGD optimizer with a learning rate of $10^{-2}$. If not specified, our default setting for local update epochs is $E = 1$, and the default setting for the amount of data at each client is $10\%$ [3] of the dataset original size. For the default non-iid setting, the FL system contains five clients. Each client exclusively owns data sampled from one of the five datasets. More details are listed in Appendix D.2.

---

[2]This data preprocessing intends to strictly control non-related factors (e.g., imbalanced sample numbers across clients), so that the experimental findings can more clearly reflect the effect of local BN. Results without truncating are reported in Appendix E.2.

[3]Choosing 10% fraction as default is based on (i) considering it as a typical setting to present general efficacy of our method; (ii) matching literature where the client size is usually around 100 to 1000 data points (McMahan et al., 2017; Li et al., 2019; Hsu et al., 2019), which is a similar scale with respect to our 10% setting (in terms of the absolute value of sample numbers).

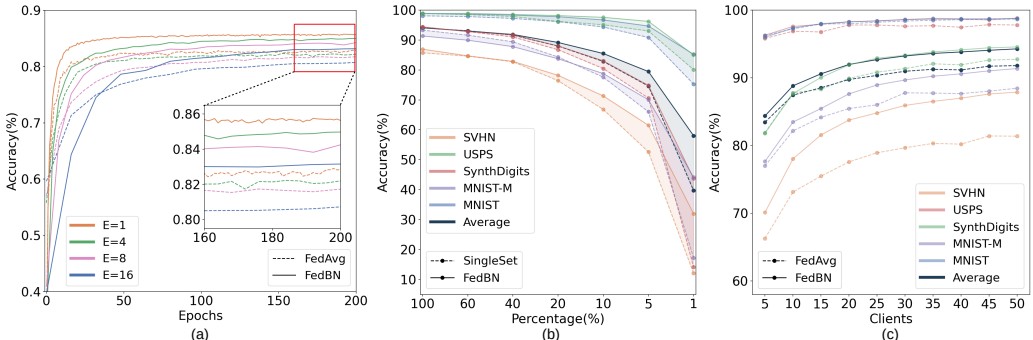

Figure 4: Analytical experimental results on: (a) Analysis on different local updating epochs. FedBN consistently outperforms FedAvg in testing accuracy. (b) Model performance over varying dataset size on local clients. (c) Testing accuracy on different levels of heterogeneity.

**Overviews:** In the following paragraphs, we present a comprehensive investigation on the properties of the proposed FedBN approach, including: (1) convergence rate; (2) behavior with respect to the choices of local update epochs; (3) performance on various amounts of data at each client; (4) effects at different level of heterogeneity; (5) comparison to state of the art (FedProx (Li et al., 2020b)), and two baselines (FedAvg and SingleSet, i.e., training an individual model within each client). In Appendix G, we also provide empirical results for including a new client with data from an unknown domain into the learning system.

**Convergence Rate:** We analyze the training loss curve of FedBN in comparison with FedAvg, as shown in Fig. 3. The loss of FedBN goes down faster and smoother than FedAvg, indicating that FedBN has a larger convergence rate. Moreover, compared to FedAvg, FedBN presents smoother and more stable loss curves during learning. These experimental observations show consensus with what given by Corollary 4.6. In addition, we present a more comprehensive comparison with different local update epochs $E$ on convergence rate of FedBN and FedAvg (see Appendix E.1). The results show similar patterns as in Fig 3.

**Analysis of Local Updating Epochs:** Aggregating at different frequencies may affect the learning behaviour. Although our theory and the default setting for the other experiment takes $E = 1$, we demonstrate FedBN is effective for cases when $E > 1$. In Fig.4 (a), we explore $E = 1, 4, 8, 16$ and compare FedBN to baseline FedAvg. As expected, an inverse relationship between the local updating epochs $E$ and testing accuracy implied for both FedBN and FedAvg shown in Fig.4 (a). Zooming into the final testing accuracy, FedBN's accuracy stably exceeds the accuracy of FedAvg on various $E$ .

**Analysis of Local Dataset Size:** We vary the data amount for each client from $100\%$ to $1\%$ of its original dataset size, in order to observe FedBN behaviour over different data capacities at each client. The results in Fig.4 (b) present the accuracy of FedBN and SingleSet [4]. Testing accuracy starts to significantly drop when each of the local client is only attributed 20% percentage of data from its original data amount. The improvement margin gained from FedBN increases as local dataset sizes decrease. The results indicate that FedBN can effectively benefit from collaborative training on distributed data, especially when each client only holds a small amount of data which are non-iid.

**Effects of Statistical Heterogeneity:** A salient question that arises is: to what degree of heterogeneity on feature shift FedBN is superior to FedAvg. To answer the question, we simulate a federated settings with varying heterogeneity as described below. We parcel each dataset into 10 subsets, one for each clients, with equal number of data samples and the same label distribution. We treat the clients generated from the same dataset as iid clients, while the clients generated from different datasets as non-iid clients.

---

[4]Detailed statistics and FedAvg results are presented in Appendix E.2.

| Method | Caltech-10 | | | | DomainNet | | | | | | ABIDE (medical) | | | |
|---|---|---|---|---|---|---|---|---|---|---|---|---|---|---|
| | A | C | D | W | C | I | P | Q | R | S | NYU | USM | UM | UCLA |
| SingleSet | 54.9 | 40.2 | 78.7 | 86.4 | 41.0 | 23.8 | 36.2 | **73.1** | 48.5 | 34.0 | 58.0 | 73.4 | 64.3 | 57.3 |
| | (1.5) | (1.6) | (1.3) | (2.4) | (0.9) | (1.2) | (2.7) | **(0.9)** | (1.9) | (1.1) | (3.3) | (2.2) | (1.4) | (2.4) |
| FedAvg | 54.1 | 44.8 | 66.9 | 85.1 | 48.8 | 24.9 | 36.5 | 56.1 | 46.3 | 36.6 | 62.7 | 73.1 | **70.7** | 64.7 |
| | (1.1) | (1.0) | (1.5) | (2.9) | (1.9) | (0.7) | (1.1) | (1.6) | (1.4) | (2.5) | (1.7) | (2.4) | **(0.5)** | (0.7) |
| FedProx | 54.2 | 44.5 | 65.0 | 84.4 | 48.9 | 24.9 | 36.6 | 54.4 | 47.8 | 36.9 | 63.3 | 73.0 | 70.5 | 64.5 |
| | (2.5) | (0.5) | (3.6) | (1.7) | (0.8) | (1.0) | (1.8) | (3.1) | (0.8) | (2.1) | (1.0) | (1.8) | (1.1) | (1.2) |
| FedBN | **63.0** | **45.3** | **83.1** | **90.5** | **51.2** | **26.8** | **41.5** | 71.3 | **54.8** | **42.1** | **65.6** | **75.1** | 68.6 | **65.5** |
| | **(1.6)** | **(1.5)** | **(2.5)** | **(2.3)** | **(1.4)** | **(0.5)** | **(1.4)** | (0.7) | **(0.8)** | **(1.3)** | **(1.1)** | **(1.4)** | (2.9) | **(1.0)** |

Table 1: We report results on three different real-world datasets with format mean(std) from 5-trial run. For Office-Caltech 10, *A, C, D ,W* are abbreviations for Amazon, Caltech, DSLR and WebCam, for DomainNet, *C, I, P, Q, R, S* are abbreviations for Clipart, Infograph, Painting, Quickdraw, Real and Sketch. For ABIDE, we list the abbreviations for the clients (i.e., medical institutions).

We start with including one client from each dataset in FL system. Then, we simultaneously add one client from each datasets while keep the existing clients $n$ times, for $n \in \{1, \ldots, 9\}$[5] . For each setting, we train models from scratch. More clients correspond to less heterogenity. We show the testing accuracy under different level of heterogeneity in Fig. 4 (c) and include a comparison with FedAvg, which is designed for iid FL. Our FedBN achieves substantially higher testing accuracy than FedAvg over all levels of heterogeneity.

**Comparison with State-of-the-art:** To further validate our method, we compare FedBN with one of the current state-of-the-art methods for non-iid FL, FedProx Li et al. (2020b), which also shares the benefit of easy adaptation to current FL frameworks in practice. We also include training on SingleSet and FedAvg as baselines. For each strategy, we split an independent testing datasets on clients and report the accuracy on the testing datasets. We perform 5-trial repeating experiment with different random seeds. The mean and standard deviation of the accuracy on each dataset over trials are shown in Fig. 5[6].

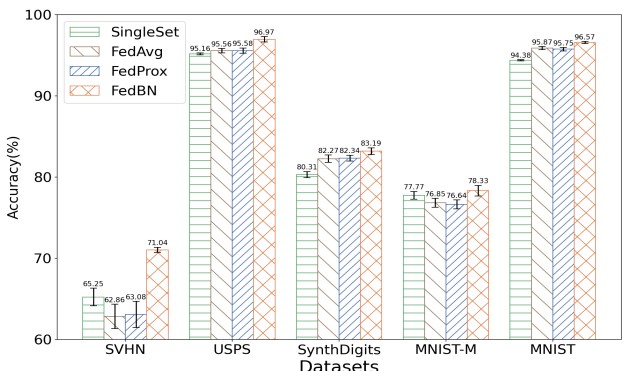

Figure 5: Performance on benchmark experiments

From the results, we can make the following observation: (1) FedBN achieves the highest accuracy, consistently outperforming the state-of-the-art and baseline methods; (2) FedBN achieves the most significant improvements on SVHN whose image appearance is very different from others (i.e., presenting more obvious feature shift); (3) FedBN shows a smaller variance in error over multiple runs, indicating its stability.

## 5.2 EXPERIMENTS ON REAL-WORLD DATASETS

To better understand how our proposed algorithm can be beneficial in real-word feature-shift non-iid, we have extensively validated the effectiveness of FedBN in comparison with other methods on three real-world datasets: image classification on Office-Caltech10 (Gong et al., 2012) with images acquired in different cameras or environments; image classification on DomainNet (Peng et al., 2019) with different image styles; and a neurodisorder diagnosis task on ABIDE I (Di Martino et al., 2014) with patients from different medical institutions[7].

---

[5]Namely, each increment contains five non-iid clients.

[6]Detailed statistics are shown in Appendix E.2

[7]More details about the datasets and training process are listed in the Appendix D.3 and D.4

**Datasets and Setup:** (1) We conduct the classification task on natural images from *Office-Caltech10*, which has four data sources composing Office-31 Saenko et al. (2010) (three data sources) and Caltech-256 datasets (one data source) Griffin et al. (2007), which are acquired using different camera devices or in different real environment with various background. Each client joining the FL system is assigned data from one of the four data sources. Thus data is non-iid across the clients. (2) Our second dataset is *DomainNet*, which contains natural images coming from six different data sources: Clipart, Infograph, Painting, Quickdraw, Real, and Sketch. Similar to (1), each client contains iid data from one of the data sources, but clients with different data sources have different feature distributions. (3) We include four medical institutions (NYU, USM, UM, UCLA; each is viewed as a client) from *ABIDE I* that collects functional brain images using different imaging equipment and protocols. We validate on a medical application for binary classification between autism spectrum disorders patients and healthy control subjects.

The Office-Caltech10 contains ten categories of objects. The DomainNet extensively contains 345 object categories and we use the top ten most common classes to form a sub-dataset for our experiments. Our classification models adopt AlexNet (Krizhevsky et al., 2012) architecture with BN added after each convolution and fully-connected layer. Before feeding into the network, all images are resized to $256 \times 256 \times 3$. For ABIDE I, each instance is represented as a 5995-dimensional vector through brain connectome computation. We use a three-layer fully connected neural network as the classifier with the hidden layers of 16 with two BN layers after the first two fully connected layers. Same as the above benchmark, we perform 5 repeated runs for each experiment.

**Results and Analysis:** The experimental results are shown in Table 1 in the form of mean (std). On Office-Caltech10, FedBN significantly outperforms the state-of-the-art method of FedProx, and improves at least $6\%$ on mean accuracy compared with all the alternative methods. On DomainNet, FedBN achieved supreme accuracy over most of the datasets. Interestingly, we find the alternative FL methods achieves comparable results with SingleSet except Quickdraw, and FedBN outperforms them over $10\%$. Surprisingly, for the above two tasks, the alternative FL strategies are ineffective in the feature shift non-iid datasets, even worse than using single client data for training for most of the clients. In ABIDE I, FedBN excell by a non-negligible margin on three clients regarding the mean testing accuracy. The results are inspiring and bring the hope of deploying FedBN to healthcare field, where data are often limited, isolated and heterogeneous on features.

## 6 CONCLUSION AND DISCUSSION

This work proposes a novel federated learning aggregation method called FedBN that keeps the local Batch Normalization parameters not synchronized with the global model, such that it mitigates feature shifts in non-IID data. We provide convergence guarantees for FedBN in realistic federated settings under the overparameterized neural networks regime, while also accounting for practical issues. In our experiments, our evaluation across a suite of federated datasets has demonstrated that FedBN can significantly improve the convergence behavior and model performance of non-IID datasets. We also demonstrate the effectiveness of FedBN in scenarios that where a new client with an unknown domain joins the FL system (see Appendix G).

FedBN is independent of the communication and aggregation strategy and thus can in practice be readily combined with different optimization algorithms, communication schemes, and aggregation techniques. The theoretical analysis of such combinations is an interesting direction for future work. We also note that since FedBN makes only lightweight modifications to FedAvg and has much flexibility to be combined with other strategies, these merits allow us to easily integrate FedBN into existing tool-kits/systems, such as Pysyft (Ryffel et al., 2018), Google TFF (Google, 2020), Flower (Beutel et al., 2020), dlplatform (Kamp & Adilova, 2020) and FedML (He et al., 2020)[8].

We believe that FedBN can improve a wide range of applications such as healthcare (Rieke et al., 2020) and autonomous driving (Kamp et al., 2018). A few interesting directions for future work include analyzing what types of differences in local data can benefit from FedBN and explore the limits of FedBN. Moreover, privacy is an essential concern in FL. Invisible BN parameters in FedBN should make attacks on local data more challenging. It would be interesting to quantify the privacy-preservation improvement in FedBN.

---

[8]The implementations on dlplatform and Flower are available, and the implementation on FedML is to appear soon.

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

APPENDIX

**Roadmap of Appendix**   The Appendix is organized as follows. We list the notations table in Section A. We provide theoretical proof of convergence in Section B. The algorithm of FedBN is described in Section C. The details of experimental setting are in Section D and additional results on benchmark datasets are in Section E. We show experiment on synthetic data in Section F. We demonstrate the ability of generalizing FedBN to test on a new client in Section G.

## A   NOTATION TABLE

| Notations | Description |
|---|---|
| $\mathbf{x}$ | features, $\mathbf{x} \in \mathbb{R}^d$ |
| $d$ | dimension of $\mathbf{x}$ |
| $y$ | labels, $y \in \mathbb{R}$ |
| $P(\cdot)$ | probability distribution |
| $N$ | total number of clients |
| $T$ | total number of epochs in training |
| $E$ | number of local iteration in FL |
| $M$ | number of training samples in each client |
| $[N]$ | set of numbers, $[N] = \{1, \ldots, N\}$ |
| $i$ | indicator for client, $i \in [N]$ |
| $j$ | indicator for sample in each client, $j \in [M]$ |
| $(\mathbf{x}_j^i, y_j^i)$ | the $j$-th training sample in client $i$ |
| $m$ | number of neurons in the first layer |
| $k$ | indicator for neuron, $k \in [m]$ |
| $\mathbf{v}_k$ | parameters for the $k$-th neuron in the first layer |
| $\| \mathbf{v} \|_{\mathbf{S}}$ | vector norm, $\| \mathbf{v} \|_{\mathbf{S}} \triangleq \sqrt{\mathbf{v}^\top \mathbf{S} \mathbf{v}}$, given a matrix $\mathbf{S}$ |
| $\mathbf{S}_i$ | covariance matrix for features in client $i$, $\mathbf{S}_i = \mathbb{E}\mathbf{x}^i \mathbf{x}^{i\top}$ |
| $p, q$ | indicator for sample, $p, q \in [NM]$ |
| $f$ | two layer ReLU neural network with BN |
| $f^*$ | two layer ReLU neural network with BN with client-specified BN parameters |
| $\mathbf{V}$ | parameters of the first phase neurons, $\mathbf{V} \in \mathbb{R}^{m \times d}$ |
| $\boldsymbol{\gamma}$ | the scaling parameter of BN |
| $\mathbf{c}$ | top layer parameters of the network |
| $\sigma(\cdot)$ | ReLU activation function, $\sigma(\cdot) = \max\{\cdot, 0\}$ |
| $N(\boldsymbol{\mu}, \boldsymbol{\Sigma})$ | Gaussian with mean $\boldsymbol{\mu}$ and covariance $\boldsymbol{\Sigma}$ |
| $U[-1, 1]$ | Rademacher distribution |
| $\alpha$ | variance of $\mathbf{v}_k$ at initialization |
| $L(f)$ | empirical risk with square loss for network $f$ |
| $\boldsymbol{\Lambda}(t)$ | evolution dynamic for FedAvg at epoch $t$ |
| $\mathbf{V}(t)$ | evolution dynamic with respect to $\mathbf{V}$ for FedAvg at epoch $t$ |
| $\mathbf{G}(t)$ | evolution dynamic with respect to $\boldsymbol{\gamma}$ for FedAvg at epoch $t$ |
| $\boldsymbol{\Lambda}^*(t)$ | evolution dynamic for FedBN at epoch $t$ |
| $\mathbf{V}^*(t)$ | evolution dynamic with respect to $\mathbf{V}$ for FedBN at epoch $t$ |
| $\mathbf{G}^*(t)$ | evolution dynamic with respect to $\boldsymbol{\gamma}$ for FedBN at epoch $t$ |
| $\lambda_{min}(A)$ | the minimal eigenvalue of matrix $A$ |
| $\mathbf{G}^\infty$ | expectation of $\mathbf{G}(t)$ |
| $\mathbf{G}^{*\infty}$ | expectation of $\mathbf{G}^*(t)$ |

Table 2: Notations occurred in the paper.

# B  CONVERGENCE PROOF

## B.1  EVOLUTION DYNAMICS

In this section, we calculate the evolution dynamics $\mathbf{\Lambda}(t)$ for training with function $f$ and $\mathbf{\Lambda}^*(t)$ for training with $f^*$. Since the parameters are updated using gradient descent, the optimization dynamics of parameters are

$$\frac{d\mathbf{v}_k}{dt} = -\frac{\partial L}{\partial \mathbf{v}_k}, \quad \frac{d\gamma_k}{dt} = -\frac{\partial L}{\partial \gamma_k}.$$

Let $f_p = f(\mathbf{x}_p^{i_p})$. Then, the dynamics of the prediction of the $p$-th data point in site $i_p$ is

$$\frac{\partial f_p}{\partial t} = \sum_{k=1}^{m} \frac{\partial f_p}{\partial \mathbf{v}_k} \frac{d\mathbf{v}_k}{dt} + \frac{\partial f_p}{\partial \gamma_k} \frac{d\gamma_k}{dt} = \underbrace{-\sum_{k=1}^{m} \frac{\partial f_p}{\partial \mathbf{v}_k} \frac{\partial L}{\partial \mathbf{v}_k}}_{T_\mathbf{v}^p} \underbrace{-\sum_{k=1}^{m} \frac{\partial f_p}{\partial \gamma_k} \frac{\partial L}{\partial \gamma_k}}_{T_\gamma^p}.$$

The gradients of $f_p$ and $L$ with respect to $\mathbf{v}_k$ and $\gamma_k$ are computed as

$$\frac{\partial f_p}{\partial \mathbf{v}_k}(t) = \frac{1}{\sqrt{m}} \frac{c_k \cdot \gamma_k(t)}{\|\mathbf{v}_k(t)\|_{\mathbf{S}_{i_p}}} \cdot \mathbf{x}_p^{\mathbf{v}_k^{i_p}(t)^\perp} \mathbb{1}_{pk}(t),$$

$$\frac{\partial L}{\partial \mathbf{v}_k}(t) = \frac{1}{\sqrt{m}} \sum_{q=1}^{NM} (f_q(t) - y_q) \frac{c_k \cdot \gamma_k(t)}{\|\mathbf{v}_k(t)\|_{\mathbf{S}_{i_q}}} \mathbf{x}_q^{\mathbf{v}_k^{i_q}(t)^\perp} \mathbb{1}_{qk}(t),$$

$$\frac{\partial f_p}{\partial \gamma_k}(t) = \frac{1}{\sqrt{m}} \frac{c_k}{\|\mathbf{v}_k(t)\|_{\mathbf{S}_{i_p}}} \sigma\left(\mathbf{v}_k(t)^\top \mathbf{x}_p\right),$$

$$\frac{\partial L}{\partial \gamma_k}(t) = \frac{1}{\sqrt{m}} \sum_{q=1}^{NM} (f_q(t) - y_q) \frac{c_k}{\|\mathbf{v}_k(t)\|_{\mathbf{S}_{i_q}}} \sigma\left(\mathbf{v}_k(t)^\top \mathbf{x}_q\right),$$

where $f_p = f(\mathbf{x}_p^{i_p})$, $\mathbf{x}_p^{\mathbf{v}_k^{i_p}(t)^\perp} \triangleq (\mathbf{I} - \frac{\mathbf{S}_{i_p}\mathbf{u}\mathbf{u}^\top}{\|\mathbf{u}\|_{\mathbf{S}_{i_p}}^2})\mathbf{x}$, and $\mathbb{1}_{pk}(t) \triangleq \mathbb{1}_{\{\mathbf{v}_k(t)^\top \mathbf{x}_p \geq 0\}}$.

We define Gram matrix $\mathbf{V}(t)$ and $\mathbf{G}(t)$ as

$$\mathbf{V}_{pq}(t) = \frac{1}{m} \sum_{k=1}^{m} (\alpha c_k \cdot \gamma_k(t))^2 \|\mathbf{v}_k(t)\|_{\mathbf{S}_{i_p}}^{-1} \|\mathbf{v}_k(t)\|_{\mathbf{S}_{i_q}}^{-1} \left\langle \mathbf{x}_p^{\mathbf{v}_k^{i_p}(t)^\perp}, \mathbf{x}_q^{\mathbf{v}_k^{i_q}(t)^\perp} \right\rangle \mathbb{1}_{pk}(t)\mathbb{1}_{qk}(t), \quad (6)$$

$$\mathbf{G}_{pq}(t) = \frac{1}{m} \sum_{k=1}^{m} c_k^2 \|\mathbf{v}_k(t)\|_{\mathbf{S}_{i_p}}^{-1} \|\mathbf{v}_k(t)\|_{\mathbf{S}_{i_q}}^{-1} \sigma\left(\mathbf{v}_k(t)^\top \mathbf{x}_p\right) \sigma\left(\mathbf{v}_k(t)^\top \mathbf{x}_q\right). \quad (7)$$

It follows that

$$T_\mathbf{v}^p(t) = \sum_{q=1}^{NM} \frac{\mathbf{V}_{pq}(t)}{\alpha^2} (f_q(t) - y_q), \quad T_\gamma^p(t) = \sum_{q=1}^{NM} \mathbf{G}_{pq}(t)(f_q(t) - y_q).$$

Let $\mathbf{f} = (f_1, \ldots, f_n)^\top = (f(\mathbf{x}_1), \ldots, f(\mathbf{x}_{NM}))^\top$. The full evolution dynamic is given by

$$\frac{d\mathbf{f}}{dt} = -\mathbf{\Lambda}(t)(\mathbf{f}(t) - \mathbf{y}), \quad \text{where} \quad \mathbf{\Lambda}(t) := \frac{\mathbf{V}(t)}{\alpha^2} + \mathbf{G}(t).$$

Similarly, we compute Gram matrix $\mathbf{V}^*(t)$ and $\mathbf{G}^*(t)$ for FedBN with $f^*$ as

$$\mathbf{V}_{pq}^*(t) = \frac{1}{m} \sum_{k=1}^{m} (\alpha c_k)^2 \gamma_{k,i_p}(t)\gamma_{k,i_q}(t) \|\mathbf{v}_k(t)\|_{\mathbf{S}_{i_p}}^{-1} \|\mathbf{v}_k(t)\|_{\mathbf{S}_{i_q}}^{-1} \left\langle \mathbf{x}_p^{\mathbf{v}_k^{i_p}(t)^\perp}, \mathbf{x}_q^{\mathbf{v}_k^{i_q}(t)^\perp} \right\rangle \mathbb{1}_{pk}(t)\mathbb{1}_{qk}(t),$$

$$\quad (8)$$

$$\mathbf{G}_{pq}^*(t) = \frac{1}{m} \sum_{k=1}^{m} c_k^2 \|\mathbf{v}_k(t)\|_{\mathbf{S}_{i_p}}^{-1} \|\mathbf{v}_k(t)\|_{\mathbf{S}_{i_q}}^{-1} \sigma\left(\mathbf{v}_k(t)^\top \mathbf{x}_p\right) \sigma\left(\mathbf{v}_k(t)^\top \mathbf{x}_q\right) \mathbb{1}\{i_p = i_q\}. \quad (9)$$

Thus, the full evolution dynamic of FedBN is

$$\frac{d\mathbf{f}^*}{dt} = -\mathbf{\Lambda}^*(t)(\mathbf{f}^*(t) - \mathbf{y}), \quad \text{where} \quad \mathbf{\Lambda}^*(t) := \frac{\mathbf{V}^*(t)}{\alpha^2} + \mathbf{G}^*(t).$$

## B.2 PROOF OF LEMMA 4.3

Dukler et al. (2020) proved that the matrix $\mathbf{G}^\infty$ is strictly positive definite. In their proof, $\mathbf{G}^\infty$ is the covariance matrix of the functionals $\phi_p$ define as

$$\phi_p(\mathbf{v}) := \sigma\left(\mathbf{v}^\top \mathbf{x}_p\right)$$

over the Hilbert space $\mathcal{V}$ of $L^2\left(N\left(0, \alpha^2 \mathbf{I}\right)\right)$. $\mathbf{G}^{*\infty}$ is strictly positive definite by showing that $\phi_1, \cdots, \phi_{NM}$ are linearly independent, which is equivalent to that

$$c_1\phi_1 + c_2\phi_2 + \cdots + c_{NM}\phi_{NM} = 0 \text{ in } \mathcal{V} \tag{10}$$

holds only for $c_p = 0$ for all $p$.

Let $\mathbf{G}_i^\infty$ denote the $i$-th $M \times M$ block matrices on the diagonal of $\mathbf{G}^\infty$. Then we have

$$\mathbf{G}^{*\infty} = diag(\mathbf{G}_1^\infty, \cdots, \mathbf{G}_N^\infty).$$

To prove that $\mathbf{G}^{*\infty}$ is strictly positive definite, we will show that $\mathbf{G}_i^\infty$ is positive definite. Let us define

$$\phi_{j,i}^*(\mathbf{v}) := \sigma\left(\mathbf{v}^\top \mathbf{x}_j\right) \mathbb{1}\{j \in \text{ site } i\}, \quad j = 1, \cdots, M.$$

Then, we are going to show that

$$c_1\phi_{1,i}^* + c_2\phi_{2,i}^* + \cdots + c_M\phi_{M,i}^* = 0 \tag{11}$$

holds only for $c_j = 0, \forall j \in [M]$. Suppose there exist $c_1, \cdots, c_M$ that are not identically 0, satisfying (11). Let the coefficients for client $i$ be $c_1, \cdots, c_M$ and let the coefficients for other client be 0. Then, we have a sequence of coefficients satisfying (10), which is a contradiction with that $\mathbf{G}^\infty$ is strictly positive definite. This implies $\mathbf{G}_i^\infty$ is strictly positive definite. Namely, $\mathbf{G}_i^\infty$'s eigenvalues are positive. Since the eigenvalues of $\mathbf{G}^{*\infty}$ are exactly the union of the eigenvalues of $\mathbf{G}_i^\infty$, $\lambda_{min}(\mathbf{G}^{*\infty})$ is positive and thus, $\mathbf{G}^{*\infty}$ is strictly positive definite.

## B.3 PROOF OF COROLLARY 4.6

To compare the convergence rates of FedAvg and FedBN when $E = 1$, we compare the exponential factor in the convergence rates, which are $(1 - \eta\mu_0/2)$ and $(1 - \eta\mu_0^*/2)$ for FedAvg and FedBN, respectively. Then, it reduces to comparing $\mu_0 = \lambda_{\min}(\mathbf{G}^\infty)$ and $\mu_0^* = \lambda_{\min}(\mathbf{G}^{*\infty})$. Comparing equation (7) and (9), $\mathbf{G}^{*\infty}$ takes the $M \times M$ block matrices on the diagonal of $\mathbf{G}^\infty$:

$$\mathbf{G}^\infty = \begin{bmatrix} \mathbf{G}_1^\infty & \mathbf{G}_{1,2}^\infty & \cdots & \mathbf{G}_{1,N}^\infty \\ \mathbf{G}_{1,2}^\infty & \mathbf{G}_2^\infty & \cdots & \mathbf{G}_{2,N}^\infty \\ \vdots & \vdots & \ddots & \vdots \\ \mathbf{G}_{1,N}^\infty & \mathbf{G}_{2,N}^\infty & \cdots & \mathbf{G}_N^\infty \end{bmatrix}, \quad \mathbf{G}^{*\infty} = \begin{bmatrix} \mathbf{G}_1^\infty & 0 & \cdots & 0 \\ 0 & \mathbf{G}_2^\infty & \cdots & 0 \\ \vdots & \vdots & \ddots & \vdots \\ 0 & 0 & \cdots & \mathbf{G}_N^\infty \end{bmatrix},$$

where $\mathbf{G}_i^\infty$ is the $i$-th $M \times M$ block matrices on the diagonal of $\mathbf{G}^\infty$. By linear algebra,

$$\lambda_{\min}(\mathbf{G}_i^\infty) \geq \lambda_{\min}(\mathbf{G}^\infty), \quad \forall i \in [N].$$

Since the eigenvalues of $\mathbf{G}^{*\infty}$ are exactly the union of eigenvalues of $\mathbf{G}_i^\infty$, we have

$$\lambda_{\min}(\mathbf{G}^{*\infty}) = \min_{i \in [N]}\{\lambda_{\min}(\mathbf{G}_i^\infty)\},$$
$$\geq \lambda_{\min}(\mathbf{G}^\infty).$$

Thus, $(1 - \eta\mu_0/2) \geq (1 - \eta\mu_0^*/2)$ and we can conclude that the convergence rate of FedBN is faster than the convergence of FedAvg.

## C    FEDBN ALGORITHM

We describe the details algorithm of our proposed FedBN as following Algorithm 1:

---

**Algorithm 1** Federated Learning using FedBN

---

**Notations:** The user indexed by $k$, neural network layer indexed by $l$, initialized model parameters: $w_{0,k}^{(l)}$, local update pace: $E$, and total optimization round $T$.

1: **for** each round $t = 1, 2, \ldots, T$ **do**
2:     **for** each user $k$ and each layer $l$ **do**
3:         $w_{t+1,k}^{(l)} \leftarrow SGD(w_{t,k}^{(l)})$
4:     **end for**
5:     **if** $\mod(t, E) = 0$ **then**
6:         **for** each user $k$ and each layer $l$ **do**
7:             **if** layer $l$ is not BatchNorm   **then**
8:                 $w_{t+1,k}^{(l)} \leftarrow \frac{1}{K} \sum_{k=1}^{K} w_{t+1,k}^{(l)}$
9:             **end if**
10:         **end for**
11:     **end if**
12: **end for**

---

# D EXPERIMENTAL DETAILS

## D.1 VISUALIZATION OF BENCHMARK DATASETS

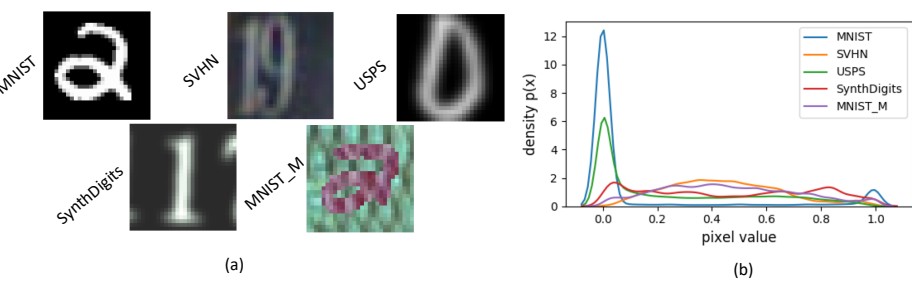

(a)                    (b)

Figure 6: Data visualization. (a) Examples from each dataset (client). (b) Non-iid feature distributions across the datasets (over random 100 samples for each dataset).

We show image examples from the five benchmark datasets and the pixel value histogram. It obviously presents the heterogeneous appearances and shifted distributions. Clients formed from the five benchmark datasets are viewed as non-iid.

## D.2 MODEL ARCHITECTURE AND TRAINING DETAILS ON BENCHMARK

We illustrate our model architecture and training details of the digits classification experiments in this section.

**Model Architecture.**  For our benchmark experiment, we use a six-layer Convolutional Neural Network (CNN) and its details are listed in Table 3.

| Layer | Details |
|---|---|
| 1 | Conv2D(3, 64, 5, 1, 2) BN(64), ReLU, MaxPool2D(2, 2) |
| 2 | Conv2D(64, 64, 5, 1, 2) BN(64), ReLU, MaxPool2D(2, 2) |
| 3 | Conv2D(64, 128, 5, 1, 2) BN(128), ReLU |
| 3 | Conv2D(64, 128, 5, 1, 2) BN(128), ReLU |
| 4 | FC(6272, 2048) BN(2048), ReLU |
| 5 | FC(2048, 512) BN(512), ReLU |
| 6 | FC(512, 10) |

Table 3: Model architecture of the benchmark experiment. For convolutional layer (Conv2D), we list parameters with sequence of input and output dimension, kernal size, stride and padding. For max pooling layer (MaxPool2D), we list kernal and stride. For fully connected layer (FC), we list input and output dimension. For BatchNormalization layer (BN), we list the channel dimension.

**Training Details.** We give detailed settings for the experiments conducted in 5.1: (1) convergence rate (Table 4), (2) analysis of local update epochs (Table 5), (3) analysis of local dataset size (Table 6), (4) effects of statistical heterogeneity (Table 7) and (5) comparison with state-of-the-art (Table 8). Each table describes the number of clients, samples and the local update epochs.

During training process, we use SGD optimizer with learning rate $10^{-2}$ and cross-entropy loss, we set batch size to 32 and training epochs to 300. For hyper-parameter $\mu$, we use the best value $\mu = 10^{-2}$ founded by grid search from the the default settings in FedProx Li et al. (2020b).

| Datasets | SVHN | USPS | SynthDigits | MNIST-M | MNIST |
|---|---|---|---|---|---|
| Number of clients | 1 | 1 | 1 | 1 | 1 |
| Number of samples | 743 | 743 | 743 | 743 | 743 |
| Local update epochs | 1 | 1 | 1 | 1 | 1 |

Table 4: Settings for convergence rate. Each dataset has 1 client with 743 samples, local update epoch is set to 1.

| Datasets | SVHN | USPS | SynthDigits | MNIST-M | MNIST |
|---|---|---|---|---|---|
| Number of clients | 1 | 1 | 1 | 1 | 1 |
| Number of samples | 743 | 743 | 743 | 743 | 743 |
| Local update epochs | 1,4,8,16 | 1,4,8,16 | 1,4,8,16 | 1,4,8,16 | 1,4,8,16 |

Table 5: Settings for local update epochs. Each dataset has 1 client with 743 samples, local update epoch for all datasets is set to 1, 4, 8, 16 successively.

| Datasets | SVHN | USPS | SynthDigits | MNIST-M | MNIST |
|---|---|---|---|---|---|
| Number of clients | 1 | 1 | 1 | 1 | 1 |
| Number of samples | $\omega$ | $\omega$ | $\omega$ | $\omega$ | $\omega$ |
| Local update epochs | 1 | 1 | 1 | 1 | 1 |

Table 6: Settings for local dataset size, we set local update epochs to 1 and each dataset has 1 client. The number of samples $\omega \in \{74, 371, 743, 1487, 2975, 4462, 7438\}$.

| Datasets | SVHN | USPS | SynthDigits | MNIST-M | MNIST |
|---|---|---|---|---|---|
| Number of clients | [1, 10] | [1, 10] | [1, 10] | [1, 10] | [1, 10] |
| Number of samples | [1, 10]×743 | [1, 10]×743 | [1, 10]×743 | [1, 10]×743 | [1, 10]×743 |
| Local update epochs | 1 | 1 | 1 | 1 | 1 |

Table 7: Settings for statistical heterogeneity, [1, 10] for the range from 1 to 10. We increase number of clients step by step and number of samples will increase accordingly.

| Datasets | SVHN | USPS | SynthDigits | MNIST-M | MNIST |
|---|---|---|---|---|---|
| Number of clients | 1 | 1 | 1 | 1 | 1 |
| Number of samples | 743 | 743 | 743 | 743 | 743 |
| Local update epochs | 1 | 1 | 1 | 1 | 1 |

Table 8: Settings for comparison with SOTA, we use 1 client with 743 samples and 1 local update epoch for comparison experiment.

### D.3 Model Architecture and Traning Details of Image Classification Task on Office-Caltech10 and DomainNet

In this section, we provide the details of our model and training process on both Office-Caltech10 Gong et al. (2012) and DomainNet Peng et al. (2019) dataset.

**Model Architecture.** For the image classification tasks on these two real-worlds datasets Office-Caltech10 and DomainNet data, we use adapted AlexNet added with BN layer after each convolutional layer and fully-connected layer (except the last layer), architecture is shown in Table 9.

| Layer | Details |
|:-----:|:-------:|
| 1 | Conv2D(3, 64, 11, 4, 2) 
 BN(64), ReLU, MaxPool2D(3, 2) |
| 2 | Conv2D(64, 192, 5, 1, 2) 
 BN(192), ReLU, MaxPool2D(3, 2) |
| 3 | Conv2D(64, 128, 5, 1, 2) 
 BN(128), ReLU |
| 3 | Conv2D(192, 384, 3, 1, 1) 
 BN(384), ReLU |
| 4 | Conv2D(384, 256, 3, 1, 1) 
 BN(256), ReLU |
| 5 | Conv2D(256, 256, 3, 1, 1) 
 BN(256), ReLU, MaxPoll2D(3, 2) |
| 6 | AdaptiveAvgPool2D(6, 6) |
| 7 | FC(9216, 4096) 
 BN(4096), ReLU |
| 8 | FC(4096, 4096) 
 BN(4096), ReLU |
| 9 | FC(4096, 10) |

Table 9: Model architecture for Office-Caltech10 and DomainNet experiment. For convolutional layer (Conv2D), we list parameters with sequence of input and output dimension, kernal size, stride and padding. For max pooling layer (MaxPool2D), we list kernal and stride. For fully connected layer (FC), we list input and output dimension. For BatchNormalization layer (BN), we list the channel dimension.

**Training Details.** Office-Caltech10 selects 10 common objects in Office-31 Saenko et al. (2010) and Caltech-256 datasets Griffin et al. (2007). There are four different data sources, one from Caltech-256 and three from Office-31, namely Amazon(images collected from online shopping website), DSLR and Webcam(images captured in office environment using Digital SLR camera and web camera).

We first reshape input images in the two dataset into $256\times256\times3$, then for training process, we use cross-entropy loss and SGD optimizer with learning rate of $10^{-2}$, batch size is set to 32 and training epochs is 300. When comparing with FedProx, we set $\mu$ to $10^{-2}$ which is tuned from the default settings. The data sample number are kept into the same size according to the smallest dataset, i.e. Office-Caltech10 uses 62 training samples and DomainNet uses 105 training samples on each dataset. In addition, for simplicity, we choose top-10 class based on data amount from DomainNet containing images over 345 categories.

### D.4 ABIDE DATASET AND TRAINING DETAILS

Here we describe the real-world medical datasets, the preprocessing and training details.

**Dataset:** The study was carried out using resting-state fMRI (rs-fMRI) data from the Autism Brain Imaging Data Exchange dataset (ABIDE I preprocessed, (Di Martino et al., 2014)). ABIDE is a consortium that provides preciously collected rs-fMRI ASD and matched controls data for the purpose of data sharing in the scientific community. We downloaded Regions of Interests (ROIs) fMRI series of the top four largest sites (UM, NYU, USM, UCLA viewed as clients) from the preprocessed ABIDE dataset with Configurable Pipeline for the Analysis of Connectomes (CPAC) and parcellated by Harvard-Oxford (HO) atlas. Skipping subjects lacking filename, resulting in 88, 167, 52, 63 subjects for UM, NYU, USM, UCLA separately. Due to a lack of sufficient data, we used sliding windows (with window size 32 and stride 1) to truncate raw time sequences of fMRI. The compositions of four sites were shown in Table 10. The number of overlapping truncate is the dataset size in a client.

|                   | NYU | UM1 | USM | UCLA1 |
|-------------------|-----|-----|-----|-------|
| Total Subject     | 167 | 88  | 52  | 63    |
| ASD Subject       | 73  | 43  | 33  | 37    |
| HC Subject        | 94  | 45  | 19  | 26    |
| ASD Percentage    | 44% | 49% | 63% | 59%   |
| fMRI Frames       | 176 | 296 | 236 | 116   |
| Overlapping Trunc | 145 | 265 | 205 | 85    |

Table 10: Data summary of the dataset used in our study.

**Training Process** : For all the strategies, we set batch size as 100. The total training local epoch is 50 with learning rate $10^{-2}$ with SGD optimizer. Local update epoch for each client is $E = 1$. We selected the best parameters $\mu = 0.2$ in FedProx through grid search.

# E    MORE EXPERIMENTAL RESULTS ON BENCHMARK DATASETS

## E.1    CONVERGENCE COMPARISON OVER FEDAVG AND FEDBN

In this section we conduct an additional convergence analysis experiment over different local update epochs settings: $E = 1, 4, 8, 16$. As shown in Fig. 7, FedBN converges faster than FedAvg under different values of $E$, which is supportive to our theoretical analysis in Section 4 and experimental results in Section 5.

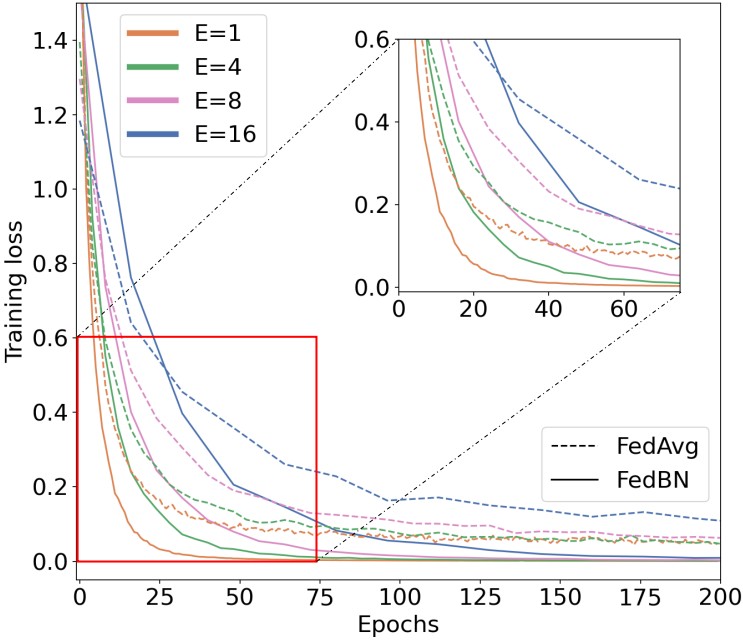

Figure 7: Training loss over epochs with different local update frequency.

## E.2    DETAILED STATISTICS OF FIGURE 5

In Figure 5, we compare the performance with respect to accuracy of FedBN and alternative methods. We show the detailed accuracy in the following Table 11.

| Methods | SVHN | USPS | Synth | MNIST-M | MNIST |
|---|---|---|---|---|---|
| Single | 65.25 (1.07) | 95.16 (0.12) | 80.31 (0.38) | 77.77 (0.47) | 94.38 (0.07) |
| FedAvg | 62.86 (1.49) | 95.56 (0.27) | 82.27 (0.44) | 76.85 (0.54) | 95.87 (0.20) |
| FedProx | 63.08 (1.62) | 95.58 (0.31) | 82.34 (0.37) | 76.64 (0.55) | 95.75 (0.21) |
| FedBN | **71.04 (0.31)** | **96.97 (0.32)** | **83.19 (0.42)** | **78.33 (0.66)** | **96.57 (0.13)** |

Table 11: The detailed statistics reported with format mean (std) of accuracy presented on Fig. 5 .

### E.3 COMPARE FEDBN WITH CENTRALIZED TRAINING

To better understand the significance of the numbers reported in our main context, we compare FedBN with centralized training, that pools all training data in to a center. We present the testing accuracy on each digit dataset in Table 12. FedBN, federated learning with data-specific BN layers, could achieve comparable performance with vanilla centralized training strategy.

| | SVHN | USPS | SynthDigits | MNIST-M | MNIST |
|---|---|---|---|---|---|
| Centralized | 74.18 (0.44) | 96.46 (0.30) | 84.57 (0.38) | 79.65 (0.24) | 96.53 (0.19) |
| FedBN | 71.04 (0.31) | 96.97 (0.32) | 83.19 (0.42) | 78.33 (0.66) | 96.57 (0.13) |

Table 12: Testing accuracy on each testing sets with format mean(std) from 5-trial run.

### E.4 DIFFERENT COMBINATIONS OF $E$ AND $B$

In this section, we show different combinations of local update epochs $E$ and batch size $B$. Specifically, $E \in \{1, 4, 16\}$ and $B \in \{10, 50, \infty\}$, $\infty$ denotes full batch learning. Following the setting in original FedAvg paper McMahan et al. (2017), we present the comparisons between FedBN and FedAvg on each combination of $E$ and $B$ in Table 13. The results are in good agreement that FedBN can consistently outperform FedAvg and robust to batch size selection. Further, we depicts the test sets accuracy vs. local epochs under different combination of $E$ and $B$ in Figure 8.

| Setting | | SVHN | USPS | SynthDigits | MNIST-M | MNIST |
|---|---|---|---|---|---|---|
| B=10, E=1 | FedAvg | 65.50 | 97.04 | 84.25 | 81.65 | 96.55 |
| | FedBN | 76.18 | 97.37 | 86.51 | 82.81 | 97.41 |
| B=10, E=4 | FedAvg | 69.80 | 96.67 | 85.63 | 82.54 | 97.21 |
| | FedBN | 76.23 | 97.04 | 86.99 | 83.14 | 97.05 |
| B=10, E=16 | FedAvg | 65.05 | 95.05 | 83.74 | 80.79 | 96.71 |
| | FedBN | 75.56 | 96.13 | 84.78 | 80.29 | 96.44 |
| B=50, E=1 | FedAvg | 62.42 | 95.32 | 81.66 | 75.28 | 96.06 |
| | FedBN | 70.70 | 97.04 | 82.74 | 78.38 | 96.57 |
| B=50, E=4 | FedAvg | 61.67 | 95.16 | 80.69 | 74.44 | 95.71 |
| | FedBN | 69.85 | 97.10 | 81.78 | 77.56 | 96.40 |
| B=50, E=16 | FedAvg | 60.00 | 94.68 | 79.37 | 73.39 | 95.28 |
| | FedBN | 67.67 | 96.94 | 80.39 | 76.54 | 95.66 |
| B=∞, E=1 | FedAvg | 60.99 | 94.57 | 79.69 | 74.36 | 95.86 |
| | FedBN | 65.98 | 96.29 | 79.75 | 76.79 | 96.15 |
| B=∞, E=4 | FedAvg | 59.07 | 95.38 | 79.88 | 73.97 | 94.51 |
| | FedBN | 65.25 | 96.34 | 79.99 | 73.96 | 95.51 |
| B=∞, E=16 | FedAvg | 61.88 | 94.68 | 78.69 | 74.36 | 95.46 |
| | FedBN | 64.39 | 95.16 | 78.22 | 73.96 | 95.57 |

Table 13: Test sets accuracy using different combinations of batch size $B$ and local update epoch $E$ on benchmark experiment with the default non-iid setting.

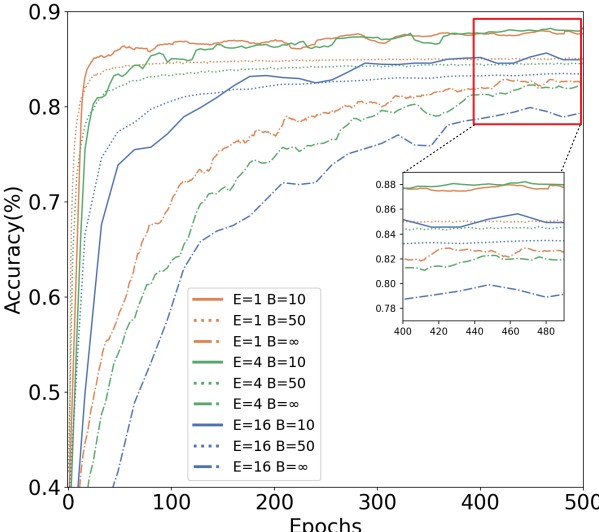

Figure 8: Test set accuracy curve (average of 5 datasets) of using different local updating epochs $E$ and batch size $B$ for FedBN.

### E.5 DETAILED STATISTICS OF VARYING LOCAL DATASET SIZE EXPERIMENT

Considering putting all results in one figure (15 lines) might affect readability of the figure, we excluded the statistics of FedAvg in our Fig. 4 (b), the ablation study of our method on the effect of local dataset size. Here, we list the full results in Table 14. It is not too surprising that at Singleset can be the best when the a local client gets a lot of data.

| | Setting | 100% | 60% | 40% | 20% | 10% | 5% | 1% |
|---|---|---|---|---|---|---|---|---|
| | SingleSet | 98.09 | 97.84 | 97.22 | 96.14 | 94.35 | 90.86 | 75.28 |
| MNIST | FedAvg | **98.96** | 98.54 | 98.13 | 97.51 | 96.22 | 93.79 | 79.94 |
| | FedBN | 98.91 | **98.63** | **98.34** | **97.78** | **96.72** | **94.67** | **85.22** |
| | SingleSet | 85.74 | 84.62 | 82.75 | 76.42 | 66.81 | 52.62 | 12.06 |
| SVHN | FedAvg | 82.08 | 79.56 | 77.37 | 70.84 | 63.80 | 49.15 | 23.67 |
| | FedBN | **86.93** | **84.72** | **82.87** | **78.20** | **71.31** | **61.53** | **31.98** |
| | SingleSet | **98.87** | 98.49 | 97.85 | 96.94 | 95.11 | 93.01 | 80.11 |
| USPS | FedAvg | 98.33 | 97.85 | 97.42 | 96.61 | 95.59 | 93.76 | 79.09 |
| | FedBN | 98.82 | **98.92** | **98.55** | **98.17** | **97.58** | **96.24** | **85.05** |
| | SingleSet | 94.33 | **92.82** | 91.02 | 86.77 | 80.47 | 70.61 | 14.10 |
| Synth | FedAvg | 93.98 | 92.57 | 91.04 | 87.03 | 82.17 | 72.76 | 42.11 |
| | FedBN | **94.40** | 92.81 | **91.75** | **88.03** | **83.06** | **74.85** | **43.76** |
| | SingleSet | **93.30** | **91.63** | **89.41** | **84.34** | 77.59 | 66.02 | 17.23 |
| MNISTM | FedAvg | 90.59 | 88.91 | 86.21 | 82.11 | 76.93 | 67.97 | 41.71 |
| | FedBN | 91.35 | 89.95 | 87.79 | 83.73 | **78.80** | **70.04** | **44.17** |

Table 14: Model performance over varying dataset sizes on local clients

### E.6 TRAINING ON UNEQUAL DATASET SIZE

In our benchmark experiment (Section 5.1), we truncate the sample size of the five datasets to their smallest number. This data preprocessing intends to strictly control non-related factors (e.g., imbalanced sample numbers across clients), so that the experimental findings can more clearly reflect the effect of local BN. In this regard, truncating datasets is a reasonable way to make each client have an equal number of data points and local update steps. It is also possible to keep the data sets in their

original size (which is unequal), by allowing clients with less data to repeat sampling. In this way, all clients use the same batch size and same local iterations of each epoch. We add results of such a setting with 10% and full original datasize in Table 15 and Table 16 respectively. It is observed that FedBN still consistently outperforms other methods.

| Method | SVHN 7943 | USPS 743 | SynthDigits 39116 | MNIST-M 5600 | MNIST 5600 |
|--------|-----------|----------|-------------------|--------------|------------|
| FedAvg | 87.00 | 98.01 | 97.55 | 88.69 | 98.75 |
| FedProx | 86.75 | 97.90 | 97.53 | 88.86 | 98.86 |
| FedBN | **89.34** | **98.28** | **97.83** | **90.34** | **98.89** |

Table 15: Testing accuracy of each clients when clients' training samples are unequal using 10% of original data. The number of training samples for each client are denoted under their names.

| Method | SVHN 79430 | USPS 7430 | SynthDigits 391160 | MNIST-M 56000 | MNIST 56000 |
|--------|------------|-----------|--------------------|--------------|-------------|
| FedAvg | 99.59 | 92.27 | 98.71 | 99.30 | 95.27 |
| FedProx | 99.50 | 92.12 | 98.66 | 99.27 | 95.44 |
| FedBN | **99.62** | **94.34** | **98.92** | **99.54** | **96.72** |

Table 16: Testing accuracy of each clients when clients' training samples are unequal using full size data. The number of training samples for each client are denoted under their names.

## F   SYNTHETIC DATA EXPERIMENT

**Settings**   We generate data from two-pair of multi-Gaussian distributions. For one pair, samples $(x, 0)$ and $(x, 1)$ are sampled from $\mathcal{N}(-1, \Sigma_1)$ and $\mathcal{N}(1, \Sigma_1)$ respectively, with coveriance $\Sigma_1 \in \mathbb{R}^{10 \times 10}$. For another pair, samples $(\widetilde{x}, 0)$ and $(\widetilde{x}, 1)$ are sampled from $\mathcal{N}(-1, \Sigma_2)$ and $\mathcal{N}(1, \Sigma_2)$ respectively, with coveriance $\Sigma_2 \in \mathbb{R}^{10 \times 10}$. Specifically, we design convariance matrix $\Sigma_1$ as an identity diagonal matrix and $\Sigma_2$ is different from $\Sigma_1$ by having non-zero values on off-diagonal entries. We train a two-layer neural network with 100 hidden neurons for 600 steps using cross-entropy loss and SGD optimizer with $1 \times 10^{-5}$ learning rate. Denote $W_k$ and $b_k$ are the in-connection weigths and bias term of neuron $k$. We initialize the model parameters with $W_k \sim \mathcal{N}(0, \alpha^2 \mathbf{I})$, $b_k \sim \mathcal{N}(0, \alpha^2)$, where $\alpha = 10$.

**Results.** The aim of synthetic experiments is to study the behavior of using FedBN with a controlled setup. We achieve  100% accuracy on binary classification for FedAvg and FedBN. Fig. 9 shows comparison of training loss curve over steps using FedAvg and FedBN, presenting that FedBN obtains significantly faster convergence than FedAvg.

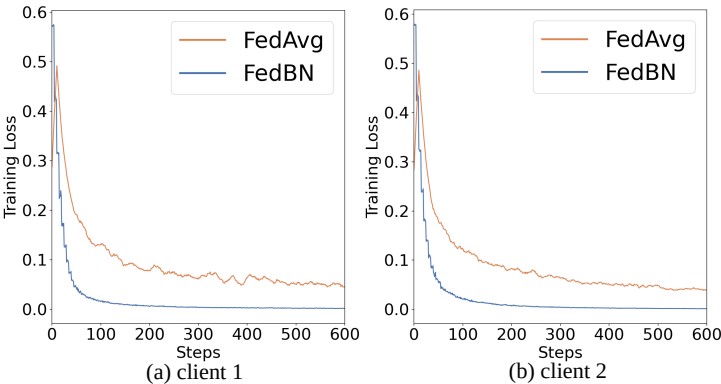

Figure 9: Training loss on synthetic data. Data in client 1 is generated from Diagonal Gaussian, client 2 is generated from combination of Diagonal Gaussian and Full Gaussian.

## G    TRANSFER LEARNING AND TESTING ON UNKNOWN DOMAIN CLIENT

In this section, we discuss out-of-domain generalization of FedBN and prove the solutions for the following two scenarios: 1) transferring FedBN to a new unknown domain clients during training; 2) testing an unknown domain client.

If a new center from another domain joins training, we can transfer the non-BN layer parameters of the global model to this new center. This new center will compute its own mean and variance statistics, and learn the corresponding local BN parameters.

Testing the global model on a new client with unknown statistics outside federation requires allowing access to local BN parameters at testing time (though BN layers are not aggregated at the global server during training). In this way, the new client can use the averaged trainable BN parameters learned at existing FL clients, and compute the (mean, variance) on its own data. Such a solution is also in line with what was done in recent literature, e.g., SiloBN (Andreux et al., 2020). We conduct the experiment with this solution for FedBN and compared its performance with FedAvg and FedProx. Specifically, we use the digits classification task and treat the two unseen datasets – Morpho-global and Morpho-local from Morpho-MNIST (Castro et al., 2019) as the two new clients. The new clients contain substantially perturbed digits. Specifically, Morpho-global containing thinning and thickening versions of MNIST digits, while Morpho-local changes MNIST by swelling and fractures. The results are listed in Table 17. It is observed that the obtained results from three methods are generally comparable in such a challenging setting, with FedBN presenting slightly higher performance on overall average accuracy.

|         | Morpho-global | Morpho-local |
|---------|---------------|--------------|
| FedBN   | **92.45**     | **94.61**    |
| FedProx | 92.35         | 94.31        |
| FedAvg  | 91.28         | 93.55        |

Table 17: Generalizing the global model to unseen-domain clients.

