# OpenReview forum: "FedBN: Federated Learning on Non-IID Features via Local Batch Normalization"
_ICLR.cc/2021/Conference — ICLR 2021 Poster_

### Official Review · AnonReviewer4 · 2020-10-26
**a new FL training strategy for non-iid data**

**Rating:** 4
**Confidence:** 4

**Review:**

This paper proposes a minor modification to the existing FL learning framework, which can be easily implemented. The major contribution of this work comes from a few theoretical analyses. The experiments are conducted on several publically accessible data sets with intuitive explanations.

Pros:
- The FL toy example is quite interesting, and we get a nice starting idea for your overall paper.
- The theoretical analysis seems pretty solid and intuitive. I didn't fully proofread all the details, but the parts that I read look correct to me.
- The experiments reveal certain properties of FedAve and FedBN, most results are convincing and the relative improvement looks pretty predictable based on the explanation.

Cons:
- The motivation on the toy example may not work well on real-world data set, especially since these assumptions may not hold in general.
- The experiments are oversimplified and lacks a proper explanation for this kind of preprocessing.
- Without any large-scale publically-accessible data set being utilized, I concern a lot about its performance on real applications.

Detailed concerns:
- The necessity of preprocessing all data set into the same setting is not provided. Why only keep 7438 examples for each data set? Does it have something to do with the non-iid setting? Is it possible to keep the data set in their original size?
- I checked the supplementary as well and didn't find how do you make sure the data set assigned to each client is non-iid.  The claimed covariate shift and concept shift didn't have corresponding experimental settings being explicitly stated, which looks quite confusing to me. Also, please add the comparison with the non-iid scenario with the one proposed in FedAvg.
-  I highly suggest the authors add a few large-scale data set to verify the effectiveness of FedBN, e.g., cifar-10, cifar-100, and imagenet. Because these data sets are more close to the scenario where each client has non-iid data.
- Each client has 10% data seems a bit too easy especially for some simple data set, e.g., MNIST, and its variants, I highly believe a comprehensive study like the one did in the FedAvg paper by changing K, B, E, and all their combinations, is the correct way to evaluate your proposed framework. Especially, FedBN is extremely easy to implement. Without these comparisons, I believe the experiment is not complete.
- Code is not attached, many details of FedBN remains unclear. The results may perform differently when we implement our own version.


Minor comments:
Inconsistent notations: sometimes FedAvg but sometimes using FedAVG.

---

> ### Author Response · Authors · 2020-11-17
> **Response to R4 - Part 1**
>
> We thank Reviewer 4 for the valuable feedback. **We would like to first clarify that the non-iid problem tackled in our work is fundamentally different from the setting in the previous FedAvg paper [1]**. FedAvg focused on label distribution differences. In contrast, we consider another important and common non-iid scenario where the data feature distribution is different across clients.
>
> > Motivation of toy example in the introduction section
>
> Thanks for recognizing the usefulness of our introductory toy example as a nice starting idea for the overall paper.  The toy example is crafted to conceptually illustrate our insights and introduce the importance of local BN layers in the case of feature shift. It may not be representative of complex real-world problems. However, we feel that increasing the realism of the example (more model parameters, a more complex feature shift) would greatly reduce the understandability of this introductory example, thus missing the point. Although the toy example is not perfectly realistic, it shares many key properties with real problems: the loss surface is highly non-convex, the learning problem is non-linear, the generalization gap varies between local minima, and the form of feature shift chosen is similar. In these regards, we think the toy example serves as a nice starting point to motivate the paper.
>
> > Clarification on data preprocessing
>
> In our original manuscript, we illustrated that “To match the setup in Section 4, we truncate the sample size of the five datasets to their smallest number, resulting in 7438 training samples in each dataset”. It is not for the non-iid setting. The non-iid setting (i.e., feature shift in our paper) roots in the five datasets themselves, generated from different distributions and/or domains.
>
> This data preprocessing intends to strictly control non-related factors (e.g., imbalanced sample numbers across clients) so that the experimental findings can more clearly reflect the effect of local BN. In this regard, truncating datasets is a reasonable way to make each client have an equal number of data points and local update steps.  It is also possible to keep the data sets in their original size, by allowing clients with fewer data to repeat sampling. In this way, all clients use the same batch size and same local iterations of each epoch. We add results of such a setting as follows and to Appendix E.6.
>
> | Methods   |  SVHN |  USPS | SynthDigits | MNIST-M | MNIST | Average |
> |-----------|:-----:|:-----:|:-----------:|:-------:|:-----:|:-------:|
> | SingleSet | 87.08 | 95.59 |    97.99    |  89.95  | 97.88 |  93.70  |
> | FedAvg    | 87.00 | 98.01 |    97.55    |  88.69  | 98.75 |  94.00  |
> | FedProx   | 86.75 | 97.90 |    97.53    |  88.86  | 98.86 |  93.98  |
> | FedBN     | 89.34 | 98.28 |    97.83    |  90.34  | 98.89 |  94.94  |
>
> It is observed that FedBN still consistently outperforms other methods. To some extent, the data preprocessing does not affect the final conclusion of this paper. But we incline to remain this preprocessing for the sake of focused and careful experimental design.
>
> > Clarification on dataset assigned to each client is non-iid in the experiment
>
> We want to clarify that the data samples assigned to each client are drawn iid from one data distribution assigned to that client. But the data across different clients are non-iid because different clients are assigned data generated from different distributions, i.e., from different datasets. In our benchmark experiment, we used five kinds of digits datasets “of different domains,” and they “have heterogeneous appearance” (see our updated visualization in Appendix D.1). In our non-iid setting, we stated that “Only one client from each of the five datasets joins the FL system. Thus the feature distributions are different across clients. That is how our benchmark experiment corresponds to the claimed covariate shift and concept shift setting.
>
> Similarly, in our real data experiment, we stated each dataset had different data sources (collected either using different devices or under different environments). Each data source was viewed as a client. Thus, our real data experiments reflect the claimed covariate shift and concept shift. To reduce confusion, we have clarified the setting in our updated version.
>
>
> [1] Mcmahan et al. Communication-Efficient Learning of Deep Networks from Decentralized Data, AISTATS. 2017

---

> ### Author Response · Authors · 2020-11-17
> **Response to R4 - Part 2**
>
> > Comparison of the non-iid scenario with the one proposed in FedAvg [1]
>
> The non-iid scenario in FedAvg -- uneven label distribution -- is fundamentally different from the non-iid scenario defined and tackled in this paper. Previous works, including FedAvg, have focused on label distribution differences or client shifts in FL. In contrast, we consider another important and common non-iid scenario where the feature distribution is different across clients. Such feature shifts can be caused by collecting data from different domains, i.e., hand-written digits in MNIST dataset v.s. street view house numbers in the SVHN dataset. In real cases, such non-iid exists in 1) images collected using different sensors, e.g., medical images acquired by different devices, 2) the same category of objects appear in different weather conditions or backgrounds, i.e., the images collected by surveillance videos and self-driving cars. We did clarify this point in the second paragraph of the Introduction and defined it in Sec. 3.
>
> > Experiments on large-scale image datasets
>
> In addition to extensive experiments on benchmark datasets, three large-scale real-world applications have been employed (Office-Caltech10, DomainNet, and a medical diagnosis task). In fact, compared to Cifar-10/100 and ImageNet, our used real-world datasets are more complex and representative for the feature shift non-IID scenario (especially with a medical task). As R1 mentioned, real-world datasets are not really common in FL and R1 appreciated seeing them in our work.
>
> Nevertheless, following your suggestion and your comments on testing on the non-iid setting in FedAvg [1], we have also conducted experiments on Cifar-10 to additionally verify the effectiveness of FedBN. The following table lists the obtained results, which still demonstrate sensible improvement over FedAvg on this public dataset. We hope this information can help address your concern about our model performance on real applications.
>
> | Methods | Accuracy |
> |---------|:--------:|
> | FedAvg  |   45.70  |
> | FedBN   |   46.14  |
>
> > Clarification on the selection of data percentage and change of K, B, E in experiments
>
> Thank you for your careful review. Besides using 10% data for each client, we have also examined a wide range of data percentages, please see Fig. 4(b). Specifically, we varied the data amount for each client in [100%,60%,40%,20%,10%,5%,1%] settings, in order to observe FedBN behavior over different data capacities at each client. In the experiments studying other parameters (e.g., K, B, E), we adopt the 10% percentage, because we consider it as a typical setting which represents the general efficacy of our method. In addition, we also notice that, in literature, client size is usually around 100~1000 data points [1,2,3], which is a similar scale w.r.t. our 10% setting (in terms of the number of data points per client). In these regards, without loss of generality, our practice in the experiment is reasonable and fair.
>
> Experiments on changing K, E are indeed included in our paper. Specifically, in Section 5 (page-6), the “Analysis of Local Updating Epochs” studies E, with results shown in Fig.4 (a) and Appendix D.4; the “Effects of Statistical Heterogeneity” studies the number of clients (K in your notation) with results shown in Fig. 4 (c).
>
> Further following your suggestion, we add experiments of varying the combination of batch size B and local updating epoch E. Please kindly check our updated Appendix E.4. The results are in good agreement that FedBN can consistently outperform FedAvg and robust to batch size selection.
>
> > Code availability
>
> We have quickly cleaned our code, benchmark datasets and pre-trained model. We have uploaded them to the anonymized link:
> https://drive.google.com/file/d/1dGAZniX_h5ArIKl4KaNEZnVOJEjUbTil/view?usp=sharing. We attach our key implementation of FedBN as below:
>
> ~~~
> def communication(server_model, models, client_weights):
>    with torch.no_grad():
>        # aggregate params
>        for key in server_model.state_dict().keys():
>            if 'bn' not in key:
>                temp = torch.zeros_like(server_model.state_dict()[key], dtype=torch.float32)
>                for client_idx in range(len(client_weights)):
>                    temp += client_weights[client_idx] * models[client_idx].state_dict()[key]
>                server_model.state_dict()[key].data.copy_(temp)
>                # update local model
>                for client_idx in range(len(client_weights)):
>                    models[client_idx].state_dict()[key].data.copy_(server_model.state_dict()[key])
>    return server_model, models
> ~~~
>
> [1] Mcmahan et al. Communication-Efficient Learning of Deep Networks from Decentralized Data, AISTATS. 2017
>
> [2] Hsu et al. 2019 Measuring the Effects of Non-Identical Data Distribution for Federated Visual Classification, arXiv preprint arXiv:1909.06335. 2019
>
> [3] Li et al. 2020 On the Convergence of FedAvg on Non-IID Data, ICLR. 2020

---

> ### Author Response · Authors · 2020-11-24
> **Appreciate your feedback on our responses**
>
> Dear Reviewer 4,
>
> As Discussion Stage 2 is about to end, we highly appreciate knowing if our responses have addressed your initial questions. We are delighted to answer your remaining concern. We appreciate your inputs and feedback very much. Thank you!
>
> Best wishes,
>
> FedBN Authors

---

### Official Review · AnonReviewer1 · 2020-10-26
**Approaching the important problem of batch normalisation in federated learning**

**Rating:** 7
**Confidence:** 5

**Review:**

This work proposes an extremely simple approach to a well-know problem within Federated Learning: batch normalisation. Indeed, FL usually imply an averaging of different model parameters trained on different distributed devices. But what happen with the running statistics that some training methods have ? Such as batch norm and some optimisers ? Well, this work proposes to address the first issue by introducing a specific batch norm method named Local Batch Normalization (FedBN). Standard BN statistics cannot simply be aggregated due to strong variations potentially occurring when training data across the clients aren't IID distributed (i.e strong shift in the input representation). To alleviate this issue, the authors propose to exclude the BN parameters from the aggregation (i.e. each client has its own BN excluded from FL).

This paper is self-contained in the sense that the initial problem is well-defined with some theoretical background and visualisations. Then a very simple solution is validated both theoretically and empirically. While the underlining idea is very, very simple, the theoretical and empirical justifications remained unclear until this work. According to this paper, we now are able to motivate the use of local BN in the context of FL. Moreover, this approach has been evaluated (convergence rate, variations w.r.t local epochs, different data distributions, heterogeneity variations, results vs SOTA)  in different non-IID FL tasks (USPS, MNIST, SynthDigits, MNIST-H). Finally, an example on "real-world" data is given to better highlight the importance of the demonstration. The proposed FedBN convincingly outperformed all the other approaches in all the setups.

However, I found the paper hard to read due to the lack of a clear plan. I think the content should be a bit more structured.

Some remarks and concerns are listed bellow:

Pros:
+ Good theoretical backup for a very simple idea.
+ Finally validating the use of local BN that was certainly used by many researchers but for "unexplained" reasons.
+ Good results compared to SOTA methods.
+ Real-world datasets are not really common in FL papers. It's great to see one.

Cons:
- The idea remains a bit "too" simple, can't we build on the top of these findings to further improve the BN ?
- Where are the appendix ?
- I find the organisation of the Sections a bit hard to follow. In particular, from the end of page 3 to the end of page 5 the reader has to follow a single big Section without any segmentation while others sections are 2 paragraphs long. As an example: Is Section 3 really needed ? Can't it be integrated into a revised Section 4 (with proper sub-sections) ?
- As BatchNorm statistics are kept local to each client. What happen at testing time ? Let's say we want to distribute the trained model to a new client that does not have already computed statistics ?

Remarks:

The paper could benefit from some proof-reading. Examples are:
- Colloray 4.6
- In particular, the average model w = (w⇤ +w⇤)/2 and average BN parameters   = ( ⇤ + ⇤)/2 has a high error -> unclear
- Widely known ag- gregation strategy in FL, FedAvg (McMahan et al., 2017) it often su↵ers when data is heterogeneous over local client. -> not correct
- As Fig. 1 shows the local squared loss between is very di↵erent between the two clients -> not correct
- What is ck in Eq 1 ?

---

> ### Author Response · Authors · 2020-11-17
> **Response to R1**
>
> We thank Reviewer 1 for the careful review and detailed summary of our paper. We have addressed all your questions in the following.
>
> > Simple idea of the proposed FedBN
>
> The proposed FedBN protocol is indeed easy to understand and implement, yet sheds new light on the setting of feature shift in FL. Moreover, the reviewers agree that the theoretical analysis is non-trivial and the experimental evaluations show substantial improvement over SOTA, both on benchmark and real-world datasets. We think that simplicity in implementation does not affect the novelty of a method. Instead, we appreciate a simple yet effective method as a strength for its significant potential of wide applicability (i.e., practical impact). Thank you for the insightful comment on further improvement, which motivates our future work to consider extensions of our FedBN based on the theoretical and empirical findings revealed in this pilot work.
>
> > Paper organization of the sections to be adjusted
>
> Thank you for the constructive suggestion. Section 3 necessarily describes fundamental concepts (e.g., non-iid setting, feature shift, FedAvg baseline) which are preliminaries to understand the context of our method. Separating it as an individual section helps to highlight this useful background information to readers, especially for those who are not very familiar with FL. Section 4 is a core part of our paper, lasting from the end of page-3 till the end of page-5. It describes the theoretical convergence analysis for FedBN in detail. Following your suggestion, we have adjusted its organization structure by segmenting this Section into three sub-sections for clearer readability. Please refer to our updated version.
>
> > How about testing the model on a new client without pre-computed statistics
>
> Thanks for the interesting and insightful point. Testing the model on a new client with unknown statistics outside the federation is somewhat out of the scope considered in this work. Nevertheless, we think that FedBN can handle such a challenging situation by allowing access to local BN parameters at the testing time (though BN layers are not aggregated at the global server during training). In this way, the new client can use the averaged (gamma, beta) of BN learned at existing FL clients, and compute the (mean, variance) on its own data. We have added an experiment with this solution for FedBN and compared its performance with FedAvg and FedProx. Specifically, we use the digits classification task and treat the two unseen datasets -- Morpho-global and Morpho-local from Morpho-MNIST [1] as the new client. The results are listed in the following table and added in Appendix G. It is observed that the obtained results from three methods are generally comparable in such a challenging setting, with FedBN presenting slightly higher performance on overall average accuracy.
>
> |                    | FedAvg  |  FedProx  |  FedBN  |
> |---------------|:----------:|:-----------:|:---------:|
> | Morpho-gloabl |  91.28 |  92.35  | 92.45 |
> | Morpho-local  |  93.55 |  94.31  | 94.61 |
>
> > Additional remarks
>
> Thanks for your careful review. We correct the typos in the updated version and thoroughly proof-read the paper.
>
> [1] Castro, Daniel C., Jeremy Tan, Bernhard Kainz, Ender Konukoglu, and Ben Glocker. "Morpho-Mnist: Quantitative assessment and diagnostics for representation learning." Journal of Machine Learning Research 20, no. 178 (2019): 1-29.

---

> > ### Comment · AnonReviewer1 · 2020-11-17
> > **Thanks for your answers.**
> >
> > Dear authors,
> >
> > Thank you for your detailed answer. My biggest concern w.r.t the "test" setup of FedBN has been properly addressed. When I discussed with different FL peoples about the findings of this work, they were mostly curious about "how to use it" at test time. We ended up with the solution proposed, but we were of course unable to validate it. Nevertheless, you did it, and it seems to work properly.
> >
> > Of course, simple does not mean bad.
> >
> > One last question ? : Any chance that the authors release the code, or implement FedBN in an existing toolkit ? (Flower, pytorch distributed ...). Honestly, this point would drastically change my view on accepting or not this work. DL is in the middle of a replication crisis, and FL suffers from that quite strongly as the setup itself is very hard to replicate ...

---

> > > ### Author Response · Authors · 2020-11-19
> > > **Thanks for your supportive feedback and code availability**
> > >
> > > Thank you very much for your quick and supportive feedback on our response. We indeed highly appreciate your in-depth thought and discussion among your surroundings about our paper. Yes, the proposed solution for test-time is currently a consensus and it yields reasonably good results. We are also interested in exploring further dedicated solutions to this important issue in our future work.
> > >
> > > We completely agree that replication is crucial in DL/FL. We have quickly cleaned our code, benchmark datasets, and pre-trained model. We have uploaded them to the anonymized link:
> > >
> > > https://drive.google.com/file/d/1kAGzRR5VEzWXh8yQD_dlS0LyC4vRDg9Q/view?usp=sharing
> > >
> > > We will further sort out our real-world datasets afterward, and we are very happy to publicly release all the implementation materials in the final version.

---

### Official Review · AnonReviewer3 · 2020-10-28
**A very good contribution**

**Rating:** 8
**Confidence:** 4

**Review:**

Update following answer:

Thanks for your detailed answer, which confirms me in my initial assessment.

------


1/ Summary of the paper

This paper introduces the use of local batch normalisation layers in order to circumvent data shifts issues in FL, called FedBN.
Building on a simplified model of BN (neural network with 1 hidden layer and BN rescaling) that was previously introduced by [1] to study the impact of BN on convergence, it is proven that the convergence of the proposed FedBN on training data can only be faster than the convergence of FedAvg on the same data points.
Parametric experiments on a simple heterogeneous dataset (built as the union of digit classification tasks) show that the proposed method yields a better performance, and a more stable one, than standard FedAvg.
Experiments on 3 real-world datasets containing heterogeneity to simulate different centers show again that the proposed FedBN method improves upon FedAvg.

2/ Acceptance decision

While ideas related to BN and FL have been floating in the community this year, this paper proposes a novel approach with extensive and convincing numerical results, and interesting theoretical results on the effect of fedBN.
Provided the paper is updated to better acknowledge other very related works, I think it should be accepted as it will be a very valuable contribution to the FL community.


3/ Supporting arguments

A/ Novelty

A very close reference not acknowledged by the paper is [2], which builds on the same ideas of domain adaptation as this paper and proposes to keep local BN weights in the FL setting, called SiloedBN, showing that results are improved and more stable with respect to FedAvg.
However, the proposed FedBN is different from siloedBN [2] in the sense that SiloedBN proposes only to keep local BN statistics, while FedBN keeps local BN layers altogether (trainable and untrainable parameters thereof).
Therefore, although not entirely new, FedBN is still novel.

B/ Theoretical results

Setting aside the drift issues encountered when the number of local updates increases, the paper uses the same formalism as [1] to get a linear convergence bound for FedAvg on the training set, which is controlled by some key constant mu_0 (the higher, the faster the convergence).
For this analysis, the contribution of this paper is to put FedBN under the same framework as [1] and to show that the resulting constant mu_0^* is larger than the constant mu_0: therefore, FedBN’s training error diminishes faster than FedAvg’s.
Although the underlying model is a simplification of the experimental reality, the problem tackled is very complex, so this result is still a significant contribution for the community.

C/ Experimental results

The experimental results are in three parts:
- An introductory toy example, which helps to better understand the importance of local BN layers in the case of domain shift.
- Benchmark on a simple hand-crafted FL digit recognition dataset built by concatenating different digit datasets with different domain shifts (e.g. MNIST, SVHN…), thereby borrowing from the domain adaptation literature. While I have some remarks to make the results even more complete (cf next section), I think these results help to better understand the behaviour of FedBN and FedAvg under heterogeneity.
- Experimental results on 3 hand-crafted FL datasets with a varying number of centers (4, 6) and different tasks. In almost all cases, FedBN significantly improves the final testing accuracy upon FedAvg and FedProx in almost centers.
These experiments are exhaustive and convincing, and are definitely an asset of this submission.

D/ Writing
Although dense, the paper is well written and easy to follow. There are minor typos.

4/ Additional comments
1. How could FedBN be adapted to transfer a model on a new center from another domain?
2. There is a minor error in the proof of Corollary 4.6, which does not invalidate the results. Indeed, the authors claim that since G^{*, \infty} = diag(G_1^{\infty}, \ldots, G_N^{\infty}), one has, for all i, \lambda_min(G^{*, \infty}) > \lambda_min(G_i^{\infty}). This is false in general: one can only claim that \lambda_min(G^{*, \infty}) \geq \min_i \lambda_min(G_i^{\infty}). Since all the minimal eigenvalues of G_i^{\infty} are lower bounded by the minimal eigenvalues of G^{\infty}, the final result still holds.
3. This is extremely minor, but in Corollary 4.6, I am not sure that in all generality one can have strict inequalities: only >= statements may be available
4. Related to the previous remark, although very interesting, the result of Corollary 4.6 is frustrating as it does not quantify the speed improvement. Even if a closed-form quanitification is out of hand, it may be interesting to produce numerical experiments on synthetic data on this simplified model to quantify the gap between the convergence bounds.
5. In section 5.1, how is computed the testing accuracy reported in figure 4? Is it the mean of the per-center accuracy on the local testing datasets? The training sets are artificially balanced, but how balanced are the testing datasets?
6. In order to better understand the significance of all the numbers reported, it would have been nice to report the performance of a model trained in a pooled-equivalent fashion, i.e. with all data points in a single center.
7. In Fig 4 b), one studies the effect of the local dataset size on the performance, and, not surprisingly, the performance of all methods diminish in this case. Why isn’t FedAvg reported here? In particular, why was the 10% fraction chosen in the other experiments, even if it led to suboptimal results?
8. In Figure 4 c), are the points with different x values related in any case? Or are they coming from independent experiments. The text « we started with including… then, we simultaneously added n clients » is a bit ambiguous.

Typos and other remarks
9. The use of both m and M in Sec 4 can be confusing
10. In Lemma 4.3, assumption 1 should be replaced by assumption 4.1
11. There are 2 typos in theorem 4.4: missing parentheses around Equation 2, and n should be replaced by N.
12. In Figure 5, the colours are hard to parse for colour-blind people.

Refs
——
[1] Dulker, Gu, and Mont\’{u}far, Optimization theory for relu neural networks trained with normalisation layers, in proceedings of ICML 2020
[2] Andreux, Ogier du Terrail, Beguier, and Tramel, Siloed federated Learning for Multi-Centric Histopathology Datasets, in proceedings of MICCAI DCL 2020

---

> ### Author Response · Authors · 2020-11-17
> **Response to R3**
>
> We thank Reviewer 3 for the very detailed and affirmative comments. We clarify your points mentioned in additional comments as follows.
>
> > How could FedBN be adapted to transfer a model on a new center from another domain?
>
> Thanks for the interesting and insightful point. If a new center from another domain joins training, we can transfer the non-BN layer parameters of the global model to this new center. This new center will compute its own mean and variance statistics, and learn the corresponding local BN parameters -- gamma and beta.
>
> > Details of the theoretical analysis and suggestion about numerical simulation
>
> Thanks for pointing this out. We have corrected typos in the updated version. Our notations in the manuscript mainly align with the notations in NKT literature (Jacot et al. 2018, Arora et al. 2019, Du et al. 2018, Dukler et al. 2020). To avoid confusion, we have provided a notation table in appendix A. We agree that a closed-form quantification will be interesting, but NTK depends on the data itself. Thus, there is no general closed-form to quantify speed improvement. The suggestion to illustrate the results using numerical simulation is great. We have added simulation results in Appendix E.
>
> > Clarification of calculation of testing accuracy reported in Fig. 4
>
> Yes, the testing accuracy is the mean of the per-center accuracy on the local testing datasets. Testing datasets are not artificially balanced, in order to keep its original nature.
>
> > Reporting results in a pooled-data setting
>
> Thanks for the good suggestion, we have added the results of a pooled-equivalent fashion. Compared with merging all data points in a single center, FedBN with local BN achieves notable higher accuracies on all datasets. We have added it in Appendix E.3.
>
> | Settings    |  SVHN |  USPS | SynthDigits | MNIST-M | MNIST | Average |
> |-------------|:-----:|:-----:|:-----------:|:-------:|:-----:|:-------:|
> | Centralized | 74.18 | 96.46 |    84.57    |  79.65  | 96.53 |  86.28  |
> | FedBN       | 76.93 | 97.69 |    87.46    |  83.57  | 97.55 |  88.64  |
>
> > Clarifications about Fig. 4 b)
>
> Fig. 4 b) is the ablation study on the local dataset size, in comparison with the baseline setting. In fact, we had the results on FedAvg. Considering putting all results in one figure (18 lines) might affect readability, we excluded it from Fig. 4 b) in the initial submission. Following your suggestion, we have included this result in Appendix E.5.  In other experiments, we chose the 10% fraction, by considering it as a typical setting to present the general efficacy of our method. We also consider that, in literature [1-3], client size is usually around 100~1000 data points, which is a similar scale w.r.t. our 10% setting (in terms of the absolute value of sample numbers).
>
> > Clarifications about Fig. 4 c)
>
> Points with different x are from independent experiments, training from scratch. We would like to clarify this setting with a concrete example. Denoting the five datasets as [a,b,c,d,e], we parcelled each dataset into 10 subsets, i.e. dataset a = {a1,a2,...,a10}, with equal number of data samples and the same label distribution. Each subsets a_i, for i in {1,...,10}, is treated as a client. The number of involved clients are progressively increased, i.e., we keep the old clients while adding new clients to FL. Specifically, for the setting of clients=5, the clients respectively use [a1,b1,c1,d1,e1], for clients=10, the clients then respectively use [a1,b1,c1,d1,e1,a2,b2,c2,d2,e2]. For each client number (i.e., x values as you mentioned), the FL model is independently trained from scratch.
>
> > Typos and other remarks
>
> Thanks for your careful review. We have corrected the minor points and cited provided valuable references in the updated version.
>
> [1] Mcmahan et al. Communication-Efficient Learning of Deep Networks from Decentralized Data, AISTATS. 2017
>
> [2] Hsu et al.  Measuring the Effects of Non-Identical Data Distribution for Federated Visual Classification, arXiv preprint arXiv:1909.06335. 2019
>
> [3] Li et al. 2020 On the Convergence of FedAvg on Non-IID Data, ICLR. 2020

---

### Official Review · AnonReviewer2 · 2020-10-30
**Review of FedBN**

**Rating:** 5
**Confidence:** 5

**Review:**

This paper develops a modified version of FedAvg by local batch normalization that is tailored for federated learning with non-i.i.d. data. Different from most of existing work that consider the unbalanced labels, this paper uses unbalanced features to motivate the non-i.i.d. federated settings. Specifically, the unbalanced features are captured by the difference in local covariances.

Merits:
1. The paper tackles an important problem in federated learning, which may have practical impact.
2. There is an effort on quantifying the performance gain of FedBN vs FedAvg.

Cons:
1. The assumption that the feature shift on local datasets is captured by the difference in local covariances needs better justification. Why this is a valid assumption?

2. The comparison is on the convergence to f and f* under FedAvg and FedBN. It is not clear if f and f* have the same learning accuracy or training loss. If not, then the comparison is not fair.

3. The convergence result is not stated in a clear form. In general, exact linear convergence to the optimal solution is not possible for SGD-type algorithm. The condition under which this holds true needs to be explicitly mentioned in Theorem 4.4 and Corollary 4.5. In addition, how the convergence depends on the averaging interval E is not clear. In FedAvg, there is non-negligible error in non-i.i.d. setting that depends on E. What technique you have used to eliminate it? These details are all unclear.

4. The current analysis is mainly following Dukler et al. (2020). The new technical challenge and difficulty in analyzing the federated version of Dukler et al. (2020) needs to be highlighted.

5. The simulations are not convincing. On one side, the state-of-the-art baseline FedAdam Reddi et al. (2020) has not been compared; on the other hand, the performance gain relative to FedAvg seems to be marginal.

---

> ### Author Response · Authors · 2020-11-19
> **Response to R2 - Part 1**
>
> We thank Reviewer 2 for your valuable feedback and patience. We are sorry for the delayed reply, as we took some time to implement the FedAdam method within our experimental framework for comparison as you suggested. We have worked hard to address your question about an empirical comparison with FedAdam which required extensive grid search experiments. We have carefully answered all your questions in the following.
>
> > Justification about the assumption of feature shift on local datasets
>
> Sorry for the confusion, the sentence “The assumption that the feature shift on local datasets is captured by the difference in local covariances” in the original submission was an intuition instead of a precise theoretical assumption. We intended to imply the importance to study the covariance differences of local datasets. The assumption is not required in the theoretical analysis, though. What we only need in Assumption 4.1 is the local covariances are not all identity matrices in the non-iid setting. We have clarified this in the updated version (cf. page 4). The feature shift in real-world datasets may be more complex beyond covariance differences. In practice, BN is performed multiple times at different layers - and thus on different representations - so that the local normalization is highly non-linear and might be able to normalize differences in local distributions that go beyond this strict definition of feature shift.
>
> > Comparison on the convergence to f and f* under FedAvg and FedBN
>
> As we know, FedAvg can lead to neural networks training divergence when clients hold data from different distributions [1,2]. The central message of our paper is that FedBN can mitigate feature shift across local datasets since local BN stabilizes training and harmonizes local loss surfaces, thereby avoiding the divergence of FedAvg.  Thus, we argue that the learning loss of FedBN should be lower than FedAvg, as illustrated in the toy example and supported by our empirical evaluation. Note also that under the NTK regime, both f and f* will converge to zero training loss. A comparison in terms of generalization bounds would be preferable, but this is highly non-trivial (even already challenging for centralized deep learning). In summary, the experiments suggest that the loss surface of FedBN leads to better learning accuracy than FedAvg and the theory proves that FedBN also converges faster. Thus, our theoretical comparison of f and f* in terms of convergence rate is insightful. The comparison supports our main message that local BN improves the convergence in federated learning.
>
> > Stating the convergence result in a clearer form
>
> Theorem 4.4 and Corollary 4.5 use gradient descent (GD) as stated in the original version of the paper. Following your suggestions to make this clearer, we have explicitly mentioned it multiple times in Section 4 in our updated manuscript.
>
> The key point in NTK is that the weights are changing slowly and the kernel is not changing much during the entire training process. The same is required in our FL analysis using NTK, so we need to bound the movements of weights for each round of synchronization. However, for E>1, controlling local dynamics becomes very challenging. To simplify tracing the optimization dynamics, we consider E=1 to make the problem tractable using NTK.  At the same time, analyzing the impact of local BN on convergence rates in other analysis frameworks (e.g., [3,4]) is also challenging.
>
> We indeed considered the impact of E in practice in our empirical evaluation. We empirically showed FedBN converges faster in Appendix E.1 Fig. 7 (old version Appendix D.2 Fig.6) and consistently achieved better accuracy and smaller error compared to FedAvg in Fig. 4(a). We observed FedBN could scale well when E increased. As expected, error increased as E increases, but still, FedBN consistently outperformed FedAvg for a wide range of E.
>
> [1] Yue Zhao, Meng Li, Liangzhen Lai, Naveen Suda, Damon Civin, and Vikas Chandra. Federated Learning with Non-IID Data, 2018
>
> [2] Brendan McMahan, Eider Moore, Daniel Ramage, Seth Hampson, and Blaise Aguera y Arcas. Communication-Efficient Learning of Deep Networks from Decentralized Data, AISTATS. 2017
>
> [3] Xiang Li, Kaixuan Huang, Wenhao Yang, Shusen Wang, and Zhihua Zhang. On the Convergence of FedAvg on Non-iid Data. ICLR 2020
>
> [4] Karimireddy, Sai Praneeth, Satyen Kale, Mehryar Mohri, Sashank J. Reddi, Sebastian U. Stich, and Ananda Theertha Suresh. SCAFFOLD: Stochastic Controlled Averaging for Federated Learning. ICML 2020

---

> ### Author Response · Authors · 2020-11-19
> **Response to R2 - Part 2**
>
> > Highlight contributions in extending Dukler et al. (2020) to FedBN
>
> Thanks for the suggestion. Generally speaking, we are working on a federated version of overparameterized two-layer ReLU networks with two kinds of BN operations (two-layer ReLU BN nets). There is a wide range of possible techniques for convergence analysis. Yet, it is non-trivial to select one for FL, especially for convergence rate comparison. Our analysis, Dukler et al. (2020), and other literature [1,2,3] are all along with the typical proof flow for NTK. Dukler et al. (2020) is a good starting point since this analysis provides a basic analytic direction of understanding how weight normalization (WN) affects deep neural network learning dynamics using NTK. However, it only offers analysis on WN and uses it in a centralized regime. Our NTK analysis focuses on the different local BN parameters and non-identical covariance matrix in FL setting.
>
> Here we summarize the challenges and our contributions :
>
> 1. How to formulate Federated Learning with BN for NTK analysis?
>
> -To our best knowledge, no existing work has formulated local BN and tackled how BN affects training in federated learning. As R1 mentioned, the theoretical and empirical justifications on BN in FL remained unclear until this work. Specifically, we write down FedBN as a two-layer ReLU BN nets with local covariance as Eq 1 and specify the initialization to fit in the NTK framework.
>
> 2. Why BN and non-iid matter in FL?
> - Given the nature of non-iid data that covariance matrices could be different and complex,  WN analysis in Dukler et al. (2020) does not directly fit in the non-identical covariance assumption in our work. In addition, BN is more commonly used in deep neural networks for reducing data shift. Therefore, we need to generalize the WN analysis to BN (cf. Theorem 4.4).
>
> 3. How to show the convergence for our Gram matrices and compare ？
> - To show the NTK convergence of FedBN, which is a keypoint, we first have to derive the specific auxiliary Gram matrix for the two-layer ReLU BN nets with local covariance which is different from Dukler et al. (2020). The positive definite property has to be shown for our Gram matrix, which defines the convergence rate. Then, we derive the bound of the kernel movements for our evolution matrices. Last, a comparison of the convergence rate is shown in our work.
>
> [1] Zeyuan Allen-Zhu, Yuanzhi Li, Zhao Song. A convergence theory for deep learning via over-parameterization. ICML 2019
>
> [2] Jan van den Brand, Binghui Peng, Zhao Song, Omri Weinstein. Training (Overparameterized) Neural Networks in Near-Linear Time. ITCS 2021
>
> [3] Simon S. Du, Xiyu Zhai, Barnabas Poczos, Aari Singh. Gradient descent provably optimizes over-parameterized neural networks. ICLR 2019

---

> ### Author Response · Authors · 2020-11-19
> **Response to R2 - Part 3**
>
> > Comparing with FedAdam Reddi et al [1]
>
> Thanks for pointing out the work. We did not compare to FedAdam [1], because our proposed approach is independent of the underlying optimization algorithm and, as such, orthogonal to FedAdam. Thus, it is possible to combine these two approaches. The approach FedAdam is interesting (though in preprint). A particular form of difference in local feature distribution is mentioned, but it is not for the general issue of feature shift. Thus, we compared FedBN to the most recent and more related state-of-the-art FedProx in handling feature shifts. We have added the line of optimization-based FL strategies in the related work section of our updated manuscript, although that is not our focus.
>
> Nevertheless, we still implement FedAdam on our benchmark experiments for a comprehensive understanding. Specifically, we follow the implementation in FedML framework (https://github.com/FedML-AI/FedML) and set client optimizers as SGD and server optimizer as Adam, which is suggested by the tensorflow implementation of FedAdam in Google TFF (https://github.com/google-research/federated/tree/master/optimization). We have performed a comprehensive hyperparameter grid search (server_lr (log10) = {1, 0.5, 0, -0.5, -1, -1.5 -2, -2.5, -3, -3.5, -4, -4.5, -5} , client_lr (log10) = {-3, -2.5, -2, -1.5, -1},  $\tau$ (log10)= {-7,-6,-5,-4,-3,-2,-1} ) to present the best possible results for FedAdam (with server_lr = 10^-3, client_lr=10^-1.5 and $\tau$ =10^-3) and FedAdam+FedBN  (with server_lr = 10^-3, client_lr=10^-1.5 and $\tau$=10^-4). Still, we cannot exclude the possibility of suboptimal parameters on our datasets.
>
> Our results are listed in the following table. We observe that FedBN outperforms FedAdam on our particular non-iid setting, and adding FedBN to FedAdam further improves the accuracy of FedAdam.
>
> |                 | MNIST |  SVHN |  USPS | Synth | MNIST-M | Average |
> |-----------------|:-----:|:-----:|:-----:|:-----:|:-------:|:-------:|
> | FedAdam         | 97.11 | 71.87 | 96.88 | 86.15 |  82.56  |  86.92  |
> | FedBN           | 97.55 | 76.93 | 97.69 | 87.46 |  83.57  |  88.64  |
> | FedAdam + FedBN | 97.43 | 76.87 | 97.53 | 87.33 |  84.51  |  88.73  |
>
> > Performance gain relative to FedAvg
>
> Thanks for the comment. We think that our empirical experiments are comprehensive and convincing with good improvement over SOTA methods, which has also been pointed out by other reviewers consistently. Notably, in results on real-world data, FedBN outperforms FedAvg by 7.2% and 5.3% with the non-iid settings on Caltech-10 datasets and DomainNet datasets (cf. Table 1).  In the benchmark experiment, the improvement is consistent within the range of 1.7-3.0% of different E (cf. Fig 4(a)). In addition, FedBN‘s average accuracy increases by 4.5% compared to FedAvg, when clients contain 1% data.
>
> [1] Sashank Reddi, Zachary Charles, Manzil Zaheer, Zachary Garrett, Keith Rush, Jakub Konečný, Sanjiv Kumar, and H. Brendan McMahan. "Adaptive Federated Optimization." arXiv preprint arXiv:2003.00295 (2020).

---

> ### Author Response · Authors · 2020-11-24
> **Appreciate your feedback on our responses**
>
> Dear Reviewer 2,
>
> As Discussion Stage 2 is about to end, we highly appreciate knowing if our responses have addressed your initial questions. We are delighted to answer your remaining concern. We appreciate your inputs and feedback very much. Thank you!
>
> Best wishes,
>
> FedBN Authors

---

### Author Response · Authors · 2020-11-19
**Thanks for your time and valuable feedback**

We thank AC and all reviewers for their time and valuable feedback. We are delighted to see that the reviewers recognize that our paper “tackles an important problem in federated learning”, proposes a “novel approach” which is “validated both theoretically and empirically” with “good results compared to SOTA”. Reviewers also highlight that our FedBN presents a “very valuable contribution to the FL community” and appreciate our effort on “real-world datasets which are not really common in FL papers” to date.

We have carefully studied all the review comments and addressed them in the following. Our paper has already been updated accordingly. We have quickly cleaned our code, benchmark datasets, and pre-trained model. We have uploaded them to the anonymized link:

https://drive.google.com/file/d/1dGAZniX_h5ArIKl4KaNEZnVOJEjUbTil/view?usp=sharing

We will further sort out our real-world datasets afterward, and we are very happy to release all materials in the final version.

---

### Decision · Program_Chairs · 2021-01-07
**Final Decision**

**Decision:**

Accept (Poster)

**Comment:**

The paper addresses the problem of batch normalization (BN) in federated learning, which is of great interest to the community including practitioners.  The proposed method here simply excludes the BN parameters from the aggregation, and evolves them locally.

As a main contribution, reviewers particularly liked the solid justification of the proposed scheme, both with substantial theory and extensive experiments. Presentation style can be slightly improved, the usage at test time can be clarified more, and some references mentioned by R3 should be added, but this overall does not affect the strong level of contributions present in the work, and the discussion phase with the authors was already constructive.